# Lossy Image Compression
# with Conditional Diffusion Models

**Ruihan Yang**
Department of Computer Science
University of California Irvine
ruihan.yang@uci.edu

**Stephan Mandt**
Department of Computer Science
University of California Irvine
mandt@uci.edu

## Abstract

This paper outlines an end-to-end optimized lossy image compression framework using diffusion generative models. The approach relies on the transform coding paradigm, where an image is mapped into a latent space for entropy coding and, from there, mapped back to the data space for reconstruction. In contrast to VAE-based neural compression, where the (mean) decoder is a deterministic neural network, our decoder is a conditional diffusion model. Our approach thus introduces an additional "content" latent variable on which the reverse diffusion process is conditioned and uses this variable to store information about the image. The remaining "texture" variables characterizing the diffusion process are synthesized at decoding time. We show that the model's performance can be tuned toward perceptual metrics of interest. Our extensive experiments involving multiple datasets and image quality assessment metrics show that our approach yields stronger reported FID scores than the GAN-based model, while also yielding competitive performance with VAE-based models in several distortion metrics. Furthermore, training the diffusion with $\mathcal{X}$-parameterization enables high-quality reconstructions in only a handful of decoding steps, greatly affecting the model's practicality. Our code is available at: `https://github.com/buggyyang/CDC_compression`

## 1   Introduction

With visual media vastly dominating consumer internet traffic, developing new efficient codecs for images and videos has become evermore crucial (Cisco, 2017). The past few years have shown considerable progress in deep learning-based image codecs that have outperformed classical codecs in terms of the inherent tradeoff between rate (expected file size) and distortion (quality loss) (Ballé et al., 2018; Minnen et al., 2018; Minnen & Singh, 2020; Zhu et al., 2021; Yang et al., 2020; Cheng et al., 2020; Yang et al., 2023). Recent research promises even more compression gains upon optimizing for perceptual quality, i.e., increasing the tolerance for imperceivable distortion for the benefit of lower

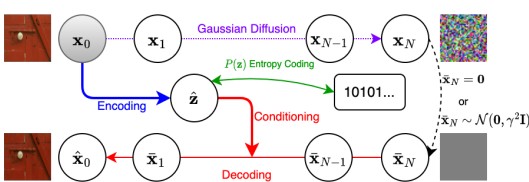

Figure 1: Overview of our proposed compression architecture. A discrete "content" latent variable $\hat{\mathbf{z}}$ contains information about the image. Upon decoding, this variable is used for conditioning a denoising diffusion process. The involved "texture" variables $\bar{\mathbf{x}}_{1:N}$ are synthesized on the fly.

rates (Blau & Michaeli, 2019). For example, adding adversarial losses (Agustsson et al., 2019; Mentzer et al., 2020) showed good perceptual quality at low bitrates.

Most state-of-the-art learned codecs currently rely on transform coding and involve hierarchical "compressive" variational autoencoders (Ballé et al., 2018; Minnen et al., 2018; Cheng et al., 2020).

37th Conference on Neural Information Processing Systems (NeurIPS 2023).

These models simultaneously transform the data into a lower dimensional latent space and use a learned prior model for entropy-coding the latent representations into short bit strings. Using either Gaussian or Laplacian decoders, these models directly optimize for low MSE/MAE distortion performance. Given the increasing focus on perceptual performance over distortion, and the fact that VAEs suffer from mode averaging behavior inducing blurriness (Zhao et al., 2017) suggest expected performance gains when replacing the Gaussian decoder with a more expressive conditional generative model.

This paper proposes to relax the typical requirement of Gaussian (or Laplacian) decoders in compression setups and presents a more expressive generative model instead: a conditional diffusion model. Diffusion models have achieved remarkable results on high-quality image generation tasks (Ho et al., 2020; Song et al., 2021b,a). By hybridizing hierarchical compressive VAEs (Ballé et al., 2018) with conditional diffusion models, we create a novel deep generative model with promising properties for perceptual image compression. This approach is related to but distinct from the recently proposed Diff-AEs (Preechakul et al., 2022), which are neither variational (as needed for entropy coding) nor tailored to the demands of image compression.

We evaluate our new compression model on four datasets and investigate a total of 16 different metrics, ranging from distortion metrics, perceptual reference metrics, and no-reference perceptual metrics. We find that the approach yields the best reported performance in FID and is otherwise comparable with the best available compression models while showing more consistent behavior across the different tasks. We also show that making the decoder more stochastic vs. deterministic offers a new possibility to steer the tradeoff between distortion and perceptual quality (Blau & Michaeli, 2019). Crucially, we find that a certain parameterization–$\mathcal{X}$-prediction (Salimans & Ho, 2022)–can yield high-quality reconstructions in only a handfull of diffusion steps.

In sum, our contributions are as follows:

- We propose a novel transform-coding-based lossy compression scheme using diffusion models. The approach uses an encoder to map images onto a contextual latent variable; this latent variable is then fed as context into a diffusion model for reconstructing the data. The approach can be modified to enhance several perceptual metrics of interest.

- We derive our model's loss function from a variational upper bound to the diffusion model's implicit rate-distortion function. The resulting distortion term is distinct from traditional VAEs in capturing a richer decoding distribution. Moreover, it achieves high-quality reconstructions in only a handful of denoising steps.

- We provide substantial empirical evidence that a variant of our approach is, in many cases, better than the GAN-based models in terms of perceptual quality, such as FID. Our base model also shows comparable rate-distortion performance with MSE-optimized baselines. To this end, we considered four test sets, three baseline models (Wang et al., 2022; Mentzer et al., 2020; Cheng et al., 2020), and up to sixteen image quality assessment metrics.

## 2 Related Work

**Transform-coding Lossy Image Compression**   The widely-established classical codecs such as JPEG (Wallace, 1991), BPG (Bellard, 2018), WEBP (Google, 2022) have recently been challenged by end-to-end learned codecs (Ballé et al., 2018; Minnen et al., 2018; Minnen & Singh, 2020; Yang et al., 2020; Cheng et al., 2020; Zhu et al., 2021; Wang et al., 2022; He et al., 2022a). These methods typically draw on the non-linear transform coding paradigm as realized by hierarchical VAEs. Usually, neural codecs are optimized to simultaneously minimize rate and *distortion* metrics, such as mean squared error or structural similarity.

In contrast to neural compression approaches targeting traditional metrics, some recent works have explored compression models to enhance *realism* (Agustsson et al., 2019; Mentzer et al., 2020; Tschannen et al., 2018; Agustsson et al., 2022; He et al., 2022b). A theoretical background for these approaches was provided by Blau & Michaeli (2019); Zhang et al. (2021); in particular Blau & Michaeli (2019) discussed the existence of a fundamental tradeoff between distortion and realism, i.e., both goals can not be achieved at the same time in a given architecture. To realize this tradeoff in compression, Mentzer et al. (2020); Agustsson et al. (2022) optimized the autoencoder-based

compression model with an additional adversarial loss, while GAN training in this context can be unstable and requires a variety of design choices.

**Diffusion Models**    Probabilistic diffusion models showed impressive performance on image generation tasks, with perceptual qualities comparable to those of highly-tuned GANs while maintaining stable training (Song & Ermon, 2019; Ho et al., 2020; Song et al., 2021b; Song & Ermon, 2019; Kingma et al., 2021; Yang et al., 2022; Ho et al., 2022; Saharia et al., 2022; Preechakul et al., 2022). Popular recent diffusion models include Dall-E2 (Ramesh et al., 2022) and Stable-Diffusion (Rombach et al., 2022). Some works also proposed diffusion models for compression. Hoogeboom et al. (2021) evaluated an autoregressive diffusion model (ADM) on a lossless compression task. Besides the difference between lossy and lossless compression, the model is only tested on low-resolution CIFAR-10 (Krizhevsky et al., 2009) dataset. Theis et al. (2022) used a generic unconditional diffusion model to lossily communicate Gaussian samples. Their method leverages "reverse channel coding", which is orthogonal to the usual transform-coding paradigm. While it's under active research, there is currently no practical method that can reduce its extensive computational cost without restrictive limitation (Li & El Gamal, 2018; Flamich et al., 2020, 2022; Theis & Ahmed, 2022).

## 3 Method

### 3.1 Background

**Denoising diffusion models**    are hierarchical latent variable models that generate data by a sequence of iterative stochastic denoising steps (Sohl-Dickstein et al., 2015; Ho et al., 2020; Song et al., 2021a; Song & Ermon, 2019). These models describe a joint distribution over data $\mathbf{x}_0$ and latent variables $\mathbf{x}_{1:N}$ such that $p_\theta(\mathbf{x}_0) = \int p_\theta(\mathbf{x}_{0:N}) d\mathbf{x}_{1:N}$. While a diffusion process (denoted by $q$) incrementally *destroys* structure, its reverse process $p_\theta$ *generates* structure. Both processes involve Markovian dynamics between a sequence of transitional steps (denoted by $n$), where

$$
\begin{aligned}
q(\mathbf{x}_n|\mathbf{x}_{n-1}) &= \mathcal{N}(\mathbf{x}_n|\sqrt{1-\beta_n}\mathbf{x}_{n-1}, \beta_n\mathbf{I}); \\
p_\theta(\mathbf{x}_{n-1}|\mathbf{x}_n) &= \mathcal{N}(\mathbf{x}_{n-1}|M_\theta(\mathbf{x}_n, n), \beta_n\mathbf{I}).
\end{aligned}
\tag{1}
$$

The variance schedule $\beta_n \in (0, 1)$ can be either fixed or learned; besides it, the diffusion process is parameter-free. The denoising process predicts the posterior mean from the diffusion process and is parameterized by a neural network $M_\theta(\mathbf{x}_n, n)$.

Denoising Diffusion Probabilistic Model (DDPM) (Ho et al., 2020) showed a tractable objective function for training the reverse process. A simplified version of their objective resulted in the following *noise parameterization*, where one seeks to predict the noise $\epsilon$ used to perturb a particular image $\mathbf{x}_0$ from the noisy image $\mathbf{x}_n$ at noise level $n$:

$$
L(\theta, \mathbf{x}_0) = \mathbb{E}_{n,\epsilon}||\epsilon - \epsilon_\theta(\mathbf{x}_n(\mathbf{x}_0), n)||^2.
\tag{2}
$$

Above, $n \sim \text{Unif}\{1, ..., N\}$, $\epsilon \sim \mathcal{N}(\mathbf{0}, \mathbf{I})$, $\mathbf{x}_n(\mathbf{x}_0) = \sqrt{\alpha_n}\mathbf{x}_0 + \sqrt{1-\alpha_n}\epsilon$, and $\alpha_n = \prod_{i=1}^{n}(1 - \beta_i)$. At test time, data can be generated by ancestral sampling using Langevin dynamics. Alternatively, Song et al. (2021a) proposed the Denoising Diffusion Implicit Model (DDIM) that follows a deterministic generation procedure after an initial stochastic draw from the prior. Our paper uses the DDIM scheme at test time, see Section 3.2 for details.

**Neural image compression**    seeks to outperform traditional image codecs by machine-learned models. Our approach draws on the transform-coding-based neural image compression approach (Theis et al., 2017; Ballé et al., 2018; Minnen et al., 2018), where the data are non-linearly transformed into a latent space, and subsequently discretized and entropy-coded under a learned "prior". The approach shows a strong formal resemblance to VAEs and shall be reviewed in this terminology.

Let $\mathbf{z}$ be a continuous latent variable and $\hat{\mathbf{z}} = \lfloor\mathbf{z}\rceil$ the corresponding rounded, integer vector. The VAE-based compression approach consists of a stochastic encoder $e(\mathbf{z}|\mathbf{x})$, a prior $p(\mathbf{z})$, and a decoder $p(\mathbf{x}|\mathbf{z})$. The model is trained using the negative modified ELBO,

$$
\mathcal{L}(\lambda, \mathbf{x}) = \mathcal{D} + \lambda\mathcal{R} = \mathbb{E}_{\mathbf{z}\sim e(\mathbf{z}|\mathbf{x})}[-\log p(\mathbf{x}|\mathbf{z}) - \lambda\log p(\mathbf{z})].
\tag{3}
$$

The first term measures the average reconstruction of the data (distortion $\mathcal{D}$, usually measured by MSE), while the second term measures the costs of entropy coding the latent variables under the prior

**Algorithm 1** Training the model (left); Encoding/Decoding data $\mathbf{x}_0$ (right). $\mathcal{X}$-prediction model.

| | |
|---|---|
| Sample $\mathbf{x}_0 \sim$ dataset | Given $N_{\text{test}}$ |
| **repeat** | $\hat{\mathbf{z}} = \lfloor \text{Enc}_\phi(\mathbf{x}_0) \rceil$ |
| $\quad n \sim \mathcal{U}(0, 1, 2, .., N_{\text{train}})$ | $\hat{\mathbf{z}} \xleftrightarrow{P(\hat{\mathbf{z}})}$ binary file (entropy code using $P(\hat{\mathbf{z}})$) |
| $\quad \epsilon \sim \mathcal{N}(\mathbf{0}, \mathbf{I})$ | $\bar{\mathbf{x}}_N = \mathbf{0}$ (or $\mathbf{x}_N \sim \mathcal{N}(\mathbf{0}, \gamma^2 \mathbf{I})$ for stochastic |
| $\quad \bar{\mathbf{x}}_n = \sqrt{\alpha_n}\mathbf{x}_0 + \sqrt{1 - \alpha_n}\epsilon$ | decoding) |
| $\quad \hat{\mathbf{z}} = \text{Enc}_\phi(\mathbf{x}_0) + \mathcal{U}(-0.5, 0.5)$ | **for** n=$N_{\text{test}}$ to 1 **do** |
| $\quad \bar{\mathbf{x}}_0 = \mathcal{X}_\theta(\bar{\mathbf{x}}_n, n/N_{\text{train}}, \hat{\mathbf{z}})$ | $\quad \epsilon_\theta = \frac{\mathbf{x}_n - \sqrt{\alpha_n}\mathcal{X}_\theta(\mathbf{x}_n(\mathbf{x}_0), \mathbf{z}, \frac{n}{N})}{\sqrt{1 - \alpha_n}}$ |
| $\quad L_{\text{D}} = \frac{\alpha_n}{1 - \alpha_n}\|\mathbf{x}_0 - \bar{\mathbf{x}}_0\|^2$ | $\quad \bar{\mathbf{x}}_0 = \mathcal{X}_\theta(\bar{\mathbf{x}}_n, n/N_{\text{test}}, \hat{\mathbf{z}})$ |
| $\quad L = (1 - \rho)L_{\text{D}} + \rho d(\bar{\mathbf{x}}_0, \mathbf{x}_0) - \lambda \log_2 P(\hat{\mathbf{z}})$ | $\quad \bar{\mathbf{x}}_{n-1} = \sqrt{\alpha_{n-1}}\bar{\mathbf{x}}_0 + \sqrt{1 - \alpha_{n-1}}\epsilon_\theta$ |
| $\quad (\theta, \phi) = (\theta, \phi) - \varepsilon \nabla_{\theta,\phi}L$ (learning rate: $\varepsilon$) | **end for** return $\bar{\mathbf{x}}_0$ |
| **until** converge | |

(bitrate $\mathcal{R}$). The encoder $e(\mathbf{z}|\mathbf{x}) = \mathcal{U}(\text{Enc}_\phi(\mathbf{x}) - \frac{1}{2}, \text{Enc}_\phi(\mathbf{x}) + \frac{1}{2})$ is a boxed-shaped distribution that simulates rounding at training time through noise injection due to the reparameterization trick. Note its differential entropy equals zero.

Once the VAE is trained, we en/decode data using the deterministic components as $\hat{\mathbf{z}} = \lfloor \text{Enc}(\mathbf{x}) \rceil$ and $\hat{\mathbf{x}} = \text{Dec}(\hat{\mathbf{z}})$. We convert the continuous $p(\mathbf{z})$ into a discrete $P(\hat{\mathbf{z}})$ using $P(\hat{\mathbf{z}}) = \text{CDF}_p(\hat{\mathbf{z}} + 0.5) - \text{CDF}_p(\hat{\mathbf{z}} - 0.5)$, where $\text{CDF}_p$ is the cumulative distribution function of $p(\mathbf{z})$; see (Yang et al., 2023) [Sec. 2.1.6] for details. The discrete prior is used for entropy coding $\hat{\mathbf{z}}$ (Ballé et al., 2018).

While VAE-based approaches have used simplistic (e.g., Gaussian) decoders, we show that can get significantly better results when defining the decoder $p(\mathbf{x}|\mathbf{z})$ as a conditional diffusion model.

## 3.2 Conditional Diffusion Model for Compression

The basis of our compression approach is a new latent variable model: the diffusion variational autoencoder. This model has a "semantic" latent variable $\mathbf{z}$ for encoding the image content, and a set of "texture" variables $\mathbf{x}_{1:N}$ describing residual information,

$$p(\mathbf{x}_{0:N}, \mathbf{z}) = p(\mathbf{x}_{0:N}|\mathbf{z})p(\mathbf{z}). \tag{4}$$

As detailed below, the decoder will follow a denoising process conditioned on $\mathbf{z}$. Drawing on methods described in Section 3.1, we use a neural encoder $e(\mathbf{z}|\mathbf{x}_0)$ to encode the image. The prior $p(\mathbf{z})$ is a two-level hierarchical prior (commonly used in learned image compression) and is used for entropy coding $\mathbf{z}$ after quantization (Ballé et al., 2018). Next, we discuss the novel decoder model.

**Decoder and training objective** We construct the conditional denoising diffusion model in a similar way to the non-variational diffusion autoencoder of Preechakul et al. (2022); Wang et al. (2023). In analogy to Eq. 1, we introduce a conditional denoising diffusion process for decoding the latent $\mathbf{z}$,

$$p_\theta(\mathbf{x}_{0:T}|\mathbf{z}) = p(\mathbf{x}_N)\prod p_\theta(\mathbf{x}_{n-1}|\mathbf{x}_n, \mathbf{z}) = p(\mathbf{x}_N)\prod \mathcal{N}(\mathbf{x}_{n-1}|M_\theta(\mathbf{x}_n, \mathbf{z}, n), \beta_n \mathbf{I}). \tag{5}$$

Since the texture latent variables $\mathbf{x}_{1:N}$ are not compressed but synthesized at decoding time, the optimal encoder and prior should be learned jointly with the decoder's marginal likelihood $p(\mathbf{x}_0|\mathbf{z}) = \int p(\mathbf{x}_{0:N}|\mathbf{z})d\mathbf{x}_{1:N}$ while targeting a certain tradeoff between rate and distortion specified by a Lagrange parameter $\lambda$. We can upper-bound this rate-distortion (R-D) objective by invoking Jensen's inequality,

$$\mathbb{E}_{\mathbf{z} \sim e(\mathbf{z}|\mathbf{x}_0)}[-\log p(\mathbf{x}_0|\mathbf{z}) - \lambda \log p(\mathbf{z})] \leq \mathbb{E}_{\mathbf{z} \sim e(\mathbf{z}|\mathbf{x}_0)}[L_{\text{upper}}(\mathbf{x}_0|\mathbf{z}) - \lambda \log p(\mathbf{z})],$$

where we introduced $L_{\text{upper}}(\mathbf{x}_0|\mathbf{z}) = -\mathbb{E}_{\mathbf{x}_{1:N} \sim q(\mathbf{x}_{1:N}|\mathbf{x}_0)}\left[\log \frac{p(\mathbf{x}_{0:N}|\mathbf{z})}{q(\mathbf{x}_{1:N}|\mathbf{x}_0)}\right]$ as the variational upper bound to the diffusion model's negative data log likelihood (Ho et al., 2020). We realize that $L_{\text{upper}}(\mathbf{x}_0|\mathbf{z})$ corresponds to a novel *image distortion* metric induced by the conditional diffusion model (in analogy to how a Gaussian decoder induces the MSE distortion). This term measures the model's ability to reconstruct the image based on $\mathbf{z}$. In contrast, $-\log p(\mathbf{z})$ measures the number of bits needed to compress $\mathbf{z}$ under the prior. As in most other works on neural image compression (Ballé

et al., 2018; Minnen et al., 2018; Yang et al., 2023), we use a box-shaped stochastic encoder $e(\mathbf{z}|\mathbf{x}_0)$ that simulates rounding by noise injection at training time.

In analogy to Eq. 2, we simplify the training objective by using the denoising score matching loss,

$$L_{\text{upper}}(\mathbf{x}_0|\mathbf{z}) \approx \mathbb{E}_{\mathbf{x}_0,n,\epsilon}||\epsilon - \epsilon_\theta(\mathbf{x}_n, \mathbf{z}, \frac{n}{N_{\text{train}}})||^2 = \mathbb{E}_{\mathbf{x}_0,n,\epsilon}\frac{\alpha_n}{1-\alpha_n}||\mathbf{x}_0 - \mathcal{X}_\theta(\mathbf{x}_n, \mathbf{z}, \frac{n}{N_{\text{train}}})||^2 \quad (6)$$

The noise level $n$ and $\alpha_n$ are defined in Eq. 2. Instead of conditioning on $n$, we condition the model on the pseudo-continuous variable $\frac{n}{N_{\text{train}}}$ which offers additional flexibility in choosing the number of denoising steps for decoding (e.g., we can use a $N_{\text{test}}$ smaller than $N_{\text{train}}$). The right-hand-side equation describes an alternative form of the loss function, where $\mathcal{X}_\theta$ directly learns to reconstruct $\mathbf{x}_0$ instead of $\epsilon$ (Salimans & Ho, 2022). We can easily derive the equivalence with $\epsilon_\theta(\mathbf{x}_n, \mathbf{z}, \frac{n}{N}) = \frac{\mathbf{x}_n - \sqrt{\alpha_n}\mathcal{X}_\theta(\mathbf{x}_n,\mathbf{z},\frac{n}{N})}{\sqrt{1-\alpha_n}}$.

**Decoding process**  Once the model is trained, we entropy-decode $\mathbf{z}$ using the prior $p(\mathbf{z})$ and conditionally decode the image $\mathbf{x}_0$ using ancestral sampling. We consider two decoding schemes: a stochastic one with $\mathbf{x}_N \sim \mathcal{N}(0, \gamma^2 I)$ (where $\gamma > 0$) and a deterministic version with $\mathbf{x}_N = \mathbf{0}$ (or $\gamma = 0$), both following the DDIM sampling method:

$$\mathbf{x}_{n-1} = \sqrt{\alpha_{n-1}}\mathcal{X}_\theta(\mathbf{x}_n, \mathbf{z}, \frac{n}{N}) + \sqrt{1-\alpha_{n-1}}\epsilon_\theta(\mathbf{x}_n, \mathbf{z}, \frac{n}{N}) \quad (7)$$

Since the variables $\mathbf{x}_{1:N}$ are not stored but generated at test time, these "texture" variables can result in variable reconstructions upon stochastic decoding (see Figure 5 for decoding with different $\gamma$). Algorithm 1 summarizes training and encoding/decoding.

**Fast decoding using $\mathcal{X}$-prediction**  In most applications of diffusion models, the iterative generative process is a major roadblock towards fast generation. Although various methods have been proposed to reduce the number of iterations, they often require additional post-processing methods, such as progress distillation (Salimans & Ho, 2022) and parallel denoising (Zheng et al., 2022).

Surprisingly, in our use case of diffusion models, we found that the $\mathcal{X}$-prediction model with *only a handfull of decoding steps* achieves comparable compression performance to the $\epsilon$-model with hundreds of steps, without the need of any post-processing. This can be explained by closely inspecting $\mathcal{X}$-prediction objective from Eq. 6 that almost looks like an autoencoder loss, with the modification that $n$ and $\mathbf{x}_n$ are given as additional inputs. When $n$ is large, $\mathbf{x}_n$ looks like pure noise and doesn't contain much information about $\mathbf{x}_0$; in this case, $\mathcal{X}$ will ignore this input and reconstruct the data based on the content latent variable $\mathbf{z}$. In contrast, if $n$ is small, $\mathbf{x}_n$ will closely resemble $\mathbf{x}_0$ and hence carry *more* information than $\mathbf{z}$ to reconstruct the image. This is to say that our diffusion objective inherits the properties of an autoencoder to reconstruct the data in a single iteration; however, successive decoding allows the model to refine this estimate and arrive at a reconstruction closer to the data manifold, with beneficial effects for perceptual properties.

**Optional Perceptual Loss**  While Eq. 6 already describes a viable loss function for our conditional diffusion compression model, we can influence the perceptual quality of the compressed images by introducing additional loss functions similar to (Mentzer et al., 2020).

First, we note that the decoded data point can be understood as a function of the low-level latent $\mathbf{x}_n$, the latent code $\mathbf{z}$, and the iteration $n$, such that $\bar{\mathbf{x}}_0 = \mathcal{X}_\theta(\mathbf{x}_n, \mathbf{z}, n/N)$ or $\frac{\mathbf{x}_n - \sqrt{1-\alpha_n}\epsilon_\theta(\mathbf{x}_n,\mathbf{z},n/N)}{\sqrt{\alpha_n}}$. When minimizing a perceptual metric $d(\cdot, \cdot)$, we can therefore add a new term to the loss:

$$L_{\text{p}} = \mathbb{E}_{\epsilon,n,\mathbf{z}\sim e(\mathbf{z}|\mathbf{x}_0)}[d(\bar{\mathbf{x}}_0, \mathbf{x}_0)] \text{ and } L_{\text{c}} = \mathbb{E}_{\mathbf{z}\sim e(\mathbf{z}|\mathbf{x}_0)}[L_{\text{upper}}(\mathbf{x}_0|\mathbf{z}) - \frac{\lambda}{1-\rho}\log p(\mathbf{z})] \quad (8)$$

$$L = \rho L_{\text{p}} + (1-\rho)L_{\text{c}}. \quad (9)$$

This loss term is weighted by an additional Lagrange multiplier $\rho \in [0, 1)$, resulting in a three-way tradeoff between rate, distortion, and perceptual quality (Yang et al., 2023).

We stress that different variations of perceptual losses for compression have been proposed (Yang et al., 2023). While this paper uses the widely-adopted LPIPS loss (Zhang et al., 2018), other approaches add an adversarial term that seek to make the reconstructions indistinguishable from reconstructed images. In this setup, Blau & Michaeli (2019) have proven mathematically that it is impossible to simultaneously obtain abitrarily good perceptual qualities and low distortions. In this paper, we observe a similar fundamental tradeoff between perception and distortion.

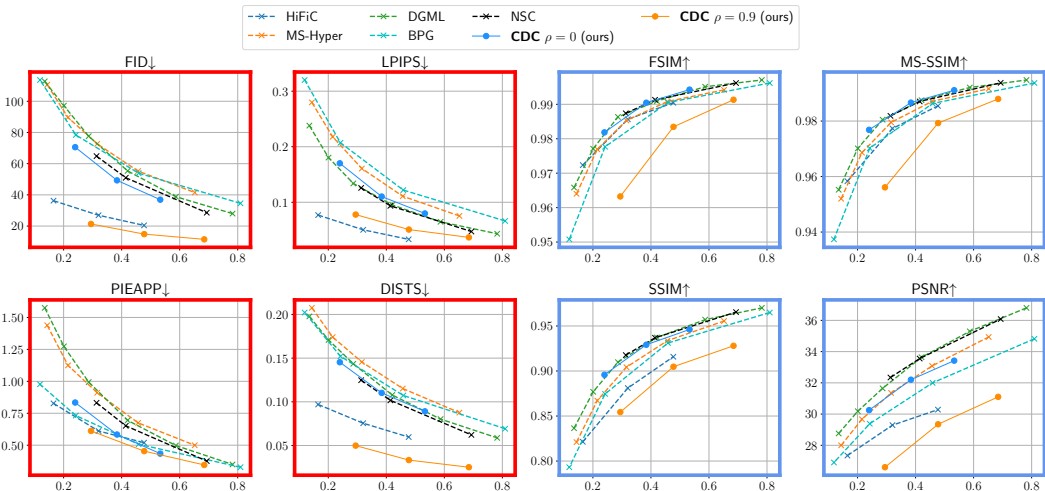

Figure 2: Tradeoffs between bitrate (x-axes, in bpp) and different metrics (y-axes) for various models tested on DIV2K. We consider both perceptual (red frames) and distortion metrics (blue frames). Arrows in the plot titles indicate whether high (↑) or low (↓) values indicate a better score. CDC (proposed) in its basic version (deterministic, without finetuning to LPIPS) compares favorably in distortion metrics, while CDC with stochastic decoding and added LPIPS losses performs favorably on perceptual metrics.

## 4 Experiments

We conducted a large-scale compression evaluation involving multiple image quality metrics and test datasets. Besides metrics measuring the traditional distortion scores, we also consider metrics that can reflect perceptual quality. While some of these metrics are fixed, others are learned from data. We will refer to our approach as "Conditional Diffusion Compression" (CDC).

**Metrics** We selected sixteen metrics for image quality evaluations, of which we present eight most widely-used ones in the main paper and the remaining eight in the appendix. Specifically, several more recently proposed learned metrics (Heusel et al., 2017; Zhang et al., 2018; Prashnani et al., 2018; Ding et al., 2020) capture perceptual properties/realism better than other non-learned methods; we denote these metrics as *perceptual metrics* and the others as *distortion metrics*. It is important to note that when calculating FID, we follow Mentzer et al. (2020) by segmenting images into non-overlapping patches of $256 \times 256$ resolution. A brief introduction about the metrics is also available in Appendix D.

**Test Data** To support our compression quality assessment, we consider the following datasets with necessary preprocessing: **1. Kodak** (Franzen, 2013): The dataset consists of 24 high-quality images at $768 \times 512$ resolution. **2. Tecnick** (Asuni & Giachetti, 2014): We use 100 natural images with $600 \times 600$ resolutions and then downsample these images to $512 \times 512$ resolution. **3. DIV2K** (Agustsson & Timofte, 2017): The validation set of this dataset contains 100 high-quality images. We resize the images with the shorter dimension being equal to 768px. Then, each image is center-cropped to a $768 \times 768$ squared shape. **4. COCO2017** (Lin et al., 2014): For this dataset, we extract all test images with resolutions higher than $512 \times 512$ and resize them to $384 \times 384$ resolution to remove compression artifacts. The resulting dataset consists of 2695 images.

**Model Training** Our model was trained using the well-established Vimeo-90k dataset (Xue et al., 2019), which includes 90,000 video clips, each containing 7-frame sequences with a resolution of $448 \times 256$ pixels, curated from vimeo.com. For each iteration, we randomly selected one frame from every video clip, which was then subject to random cropping to achieve a uniform resolution of $256 \times 256$ pixels.

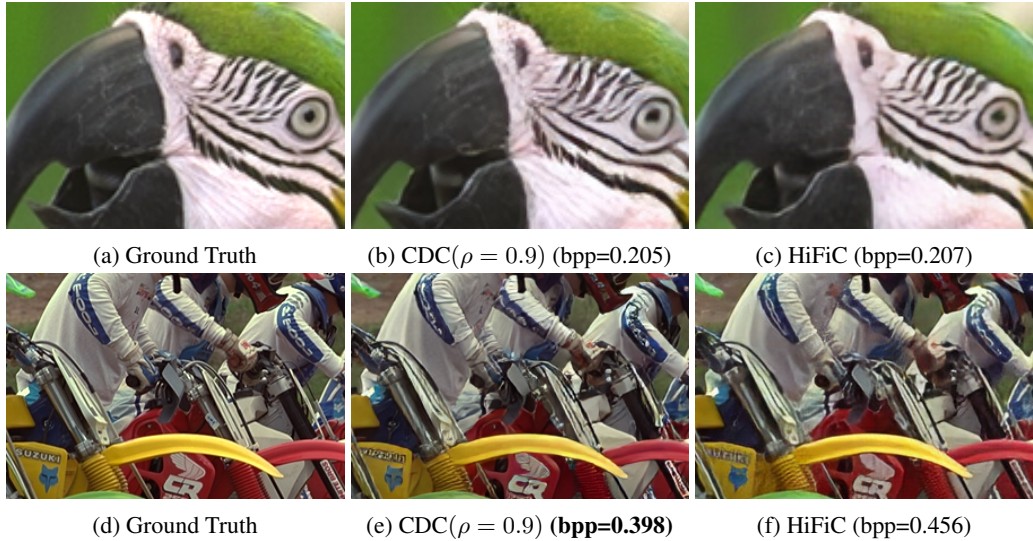

Figure 3: Reconstructed Kodak images (cropped images, see full images in Appendix F). 1st row: compared to HiFiC under similar bitrate, our model retains more details around the eyes of the parrot. 2nd row: our model still gets slightly better visual reconstruction than HiFiC while using *less* bitrate.

The training procedure initiated with a warm-up phase, setting $\lambda$ to $10^{-6}$ and running the model for approximately 700,000 steps. Subsequently, we increased $\lambda$ to align with the desired bitrates and continued training for an additional 1,000,000 steps until the model reached convergence.

For the $\epsilon$-prediction model, our training utilized diffusion process comprising $N_{\text{train}} = 20,000$ steps. Conversely, the number of diffusion steps for the $\mathcal{X}$-prediction model is $N_{\text{train}} = 8,193$. We implemented a linear variance schedule to optimize the $\epsilon$-prediction model, while a cosine schedule was selected for the $\mathcal{X}$-prediction model optimization. Throughout the training regime, we maintained a batch size of 4. The Adam optimizer (Kingma & Ba, 2014) was employed to facilitate efficient convergence. We commenced with an initial learning rate of $lr = 5 \times 10^{-5}$, which was reduced by 20% after every 100,000 steps, ultimately clipped to a learning rate of $lr = 2 \times 10^{-5}$.

### 4.1 Baseline Comparisons

**Baselines and Model Variants** We showed two variants of our $\mathcal{X}$-prediction CDC model. Our first proposed model is optimized in the presence of an additive perceptual reconstruction term at $\rho = 0.9$, which is the largest $\rho$-value we chose. For this variant, we used $\mathbf{x}_N \sim \mathcal{N}(0, \gamma^2 I)$ with $\gamma = 0.8$ to reconstruct the images. The other proposed version is the base model, trained without the additional perceptual term ($\rho = 0$) and using a deterministic decoding with $\mathbf{x}_N = 0$. As discussed below, this base version performs better in terms of distortion metrics, while the stochastic and LPIPS-informed version performs better in perceptual metrics.

We compare our method with several neural compression methods. The best reported perceptual results were obtained by **HiFiC** (Mentzer et al., 2020). This model is optimized by an adversarial network and employs additional perceptual and traditional distortion losses (LPIPS and MSE). In terms of rate-distortion performance, two VAE models are selected: **DGML** (Cheng et al., 2020) and **NSC** (Wang et al., 2022). Both are the improvements over the MSE-trained Mean-Scale Hyperprior (**MS-Hyper**) architecture (Minnen et al., 2018). For a fair comparison, we did not use the content-adaptive encoding for NSC model. For comparisons with classical codecs, we choose **BPG** as a reference.

Figure 2 illustrates the tradeoff between bitrate and image quality using the DIV2K dataset. We only employ **17** steps to decode the images with $\mathcal{X}$-prediction model, which is much more efficient than $\epsilon$-prediction model that requires hundreds of steps to achieve comparable performance (see Appendix H for results on other datasets and comparison with $\epsilon$-prediction). In the figure, dashed lines represent the baseline models, while solid lines depict our proposed CDC models. The eight shown plots present two different types of metrics, distinguished by their respective frame colors.

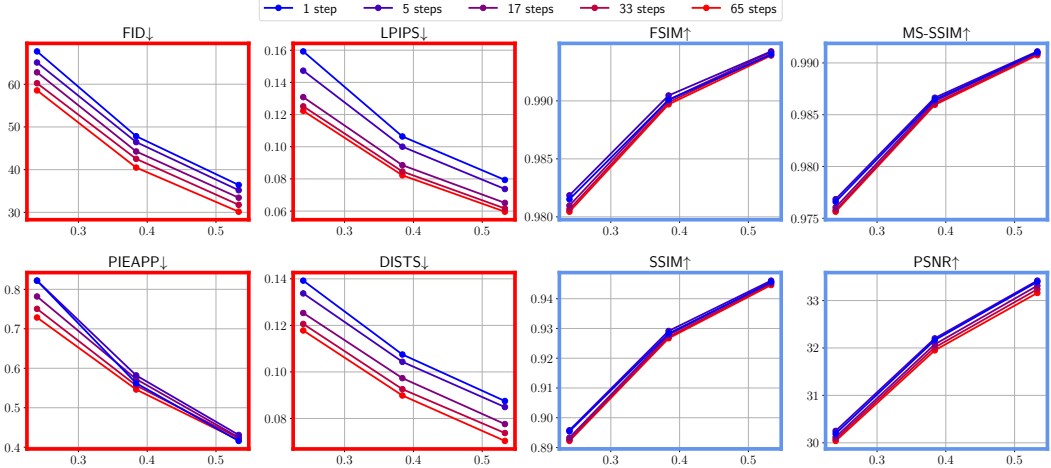

Figure 4: Compression performance with different numbers of decoding step. We use $\gamma = 0$ (deterministic decoding) to plot distortion curves and $\gamma = 1$ for perceptual quality curves.

- **Perceptual Metrics (red)**. The red subplots depict perceptual metrics that assess the compression for realism performance. Our findings reveal that our proposed CDC model ($\rho = 0.9$) achieves the best performance in three out of four metrics. The closest competitor is the HiFiC baseline. Notably, HiFiC demonstrates the highest score in LPIPS but exhibits suboptimal performance in all other metrics.

- **Distortion Metrics (blue)**. The blue subfigures present distortion-based metrics. We note that the CDC model with $\rho = 0$ produces relatively favorable results in distortion metrics, excluding PSNR. It shows on-par scores with the best baselines in FSIM, SSIM, and MS-SSIM scores, despite none of the shown models being specifically optimized for these three metrics. In contrast, "classical" neural compression models (Minnen et al., 2018; Wang et al., 2022; Cheng et al., 2020) directly target MSE distortion by minimizing an ELBO objective with Gaussian decoders, resulting in better PSNR scores.

Our proposed versions CDC ($\rho$=0) and CDC ($\rho$=0.9) show qualitative differences in perceptual and distortion metrics. Setting $\rho = 0$ only optimizes a trade-off between bitrate and the diffusion loss; compared to $\rho = 0.9$, this results in better performance in model-based distortion metrics (i.e., except PSNR). Fig. 3 qualitatively shows that the resulting decoded images show fewer over-smoothing artifacts than VAE-based codecs (Cheng et al., 2020). In contrast, CDC ($\rho$=0.9) performs the best in perceptual metrics. These are often based on extracted neural network features, such as Inception or VGG (Szegedy et al., 2016; Simonyan & Zisserman, 2014), and are more susceptible to image realism. By varying $\rho$, we can hence control a three-way trade-off among distortion, perceptual quality, and bitrate (See Appendix E for results with other $\rho$ values).

**Distortion vs. Perception** Our experiments revealed the aforementioned distortion-perception tradeoff in learned compression (Blau & Michaeli, 2019). In contrast to perceptual metrics, distortions such as PSNR are very sensitive to imperceptible image translations (Wang et al., 2005). The benefit of distortions is that they carry out a direct comparison between the reconstructed and original image, albeit using a debatable metric (Dosovitskiy & Brox, 2016). The question of whether distortion or perceptual quality is more relevant may ultimately not be easily solvable; yet it is plausible that most compression gains can be expected when targeting a combination of perception/realism and distortion, rather than distortion alone (Mentzer et al., 2020; Yang et al., 2023). Especially, our method's strong performance in terms of FID, one of the most widely-adopted perceptual evaluation schemes (Ho et al., 2020; Song & Ermon, 2019; Mentzer et al., 2020; Brock et al., 2019; Song et al., 2021a,b), seems promising in this regard.

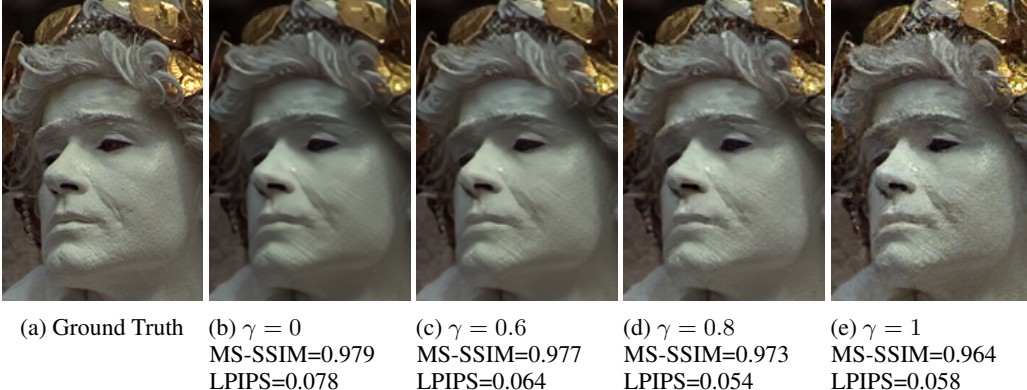

(a) Ground Truth    (b) $\gamma = 0$    (c) $\gamma = 0.6$    (d) $\gamma = 0.8$    (e) $\gamma = 1$
MS-SSIM=0.979    MS-SSIM=0.977    MS-SSIM=0.973    MS-SSIM=0.964
LPIPS=0.078    LPIPS=0.064    LPIPS=0.054    LPIPS=0.058

Figure 5: Qualitative comparison of deterministic and stochastic decoding methods. Deterministic decoding typically results in a smoother image reconstruction. By increasing the noise $\gamma$ used upon decoding the images, we observe more and more detail and rugged texture on the face of the sculpture. ($\gamma = 0.8$) show the best agreement with the ground truth image. All the images share the same bpp.

## 4.2   Ablation Studies

**Influence of decoding steps**    In the previous section, we demonstrated that the CDC $\mathcal{X}$-prediction model can achieve decent performance with a small number of decoding steps. In Figure 4, we further investigate the compression performance of the $\mathcal{X}$-prediction($\rho = 0$) model using different decoding steps. Our findings reveal that when employing stochastic decoding, the model consistently produces better *perceptual* results as the number of decoding steps increases. However, in the case of deterministic decoding, more decoding steps do not lead to a substantial improvement in *distortion*.

Our findings show that the $\mathcal{X}$-prediction model can behave similarly to a Gaussian VAE decoder. In this scenario, the latent code $\mathbf{z}$ becomes the primary determinant of the decoding outcome, enabling the model to reconstruct the original image $\mathbf{x}_0$ with a single decoding step. However, even a single decoding step has a tendency to reconstruct data closer to the data mode, only guaranteeing an acceptable distortion score. To effectively improve perceptual quality, it is crucial to incorporate more iterative decoding steps, particularly when utilizing stochastic decoding. Thus, we further explore the impact of stochastic decoding through the following ablation experiment.

**Stochastic Decoding**    By adjusting the noise level parameter, denoted as $\gamma$, during the image decoding process, we can achieve different decoding outcomes. In order to investigate the impact of the noise on the decoding results, we present Figure 5, which provides both quantitative and qualitative evidence for 4 candidates $\gamma$ values. Our findings indicate that larger values lead to improved perceptual quality and higher distortion, as evidenced by lower LPIPS and lower MS-SSIM. Values of $\gamma$ greater than 0.8 not only increase distortion but also diminish perceptual quality. In terms of finding the optimal balance, a $\gamma$ value of 0.8 offers the lowest LPIPS and the best qualitative outcomes as shown in Figure 5. From a qualitative standpoint, we notice that the noise introduces plausible high-frequency textures. Although these textures may not perfectly match the uncompressed ones (which is impossible), they are visually appealing when an appropriate $\gamma$ is chosen. For additional insights into our decoding process, we provide visualizations of the decoding steps in Appendix G. These visualizations showcase how "texture" variables evolve during decoding. We observe that this phenomenon becomes particularly pronounced when employing the $\epsilon$-prediction, as $\mathcal{X}$-prediction models exhibit a stronger inclination towards reconstruction.

## 5   Conclusion & Discussion

We proposed a transform-coding-based neural image compression approach using diffusion models. We use a denoising decoder to iteratively reconstruct a compressed image encoded by an ordinary neural encoder. Our loss term is derived from first principles and combines rate-distortion variational autoencoders with denoising diffusion models. We conduct quantitative and qualitative experiments to compare our method against several GAN and VAE based neural codecs. Our approach achieves

promising results in terms of the rate-perception tradeoff, outperforming GAN baselines in three out of four metrics, including FID. In terms of classical distortion, our approach still performs comparable to highly-competitive baselines.

**Limitations & Societal Impacts** In our quest to enhance compression performance, further improvement can be achieved by integrating advanced techniques such as autoregressive entropy models or iterative encoding. We leave such studies for future research.

One critical societal concern associated with neural compression, particularly when prioritizing perceptual quality, is the risk of misrepresenting data in the low bitrate regime. Within this context, there is a possibility that the model may generate hallucinated lower-level details. For example, compressing a human face may lead to a misrepresentation of the person's identity, which raises important considerations regarding fairness and trustworthiness in learned compression.

## 6 Acknowledgement

The authors acknowledge support by the IARPA WRIVA program, the National Science Foundation (NSF) under the NSF CAREER Award 2047418; NSF Grants 2003237 and 2007719, the Department of Energy, Office of Science under grant DE-SC0022331, as well as gifts from Intel, Disney, and Qualcomm.

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

## A Pretrained Baselines

We refer to Bégaint et al. (2020) for pretrained MS-Hyper and DGML models. For HiFiC model, we use the pretrained models implemented in the publicly available repositories[1]. Both models were sufficiently trained on natural image datasets (Xue et al., 2019; Kuznetsova et al., 2020). For NSC (Wang et al., 2022) baseline, we use the official codebase[2] and DIV2k training dataset to train the model.

## B Architectures

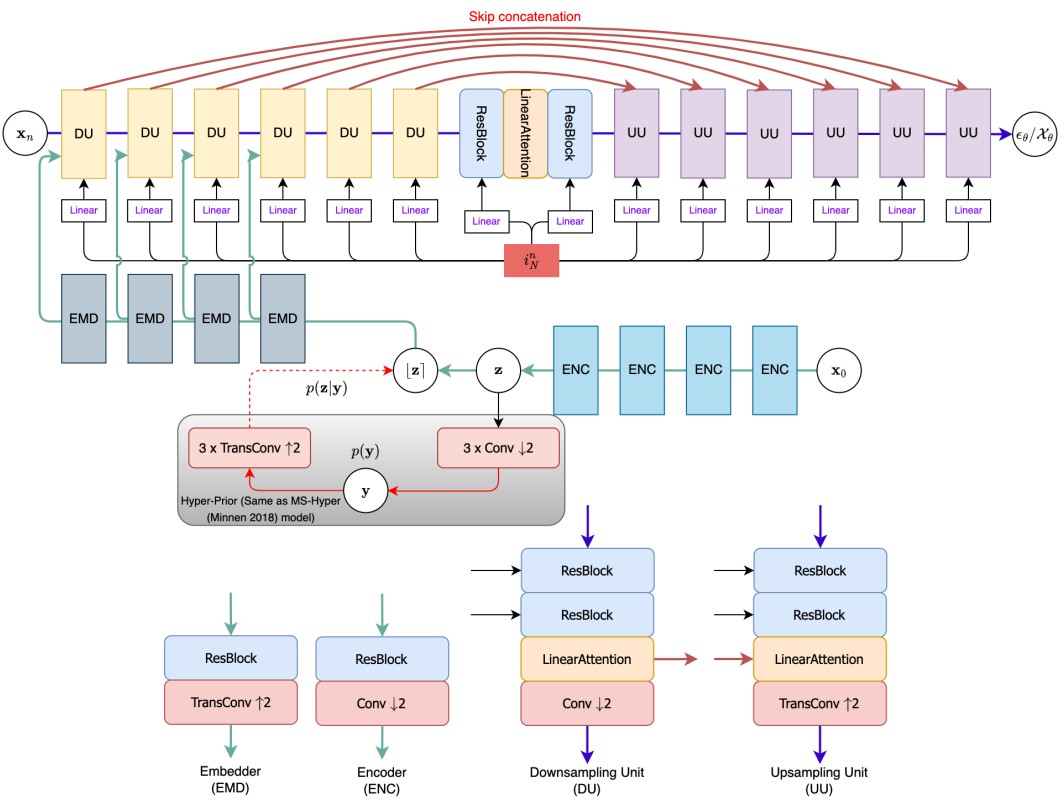

Figure 6: Visualization of our model architecture

The denoising module's design adopts a U-Net architecture similar to that of the DDIM (Song et al., 2021a) and DDPM (Ho et al., 2020) projects. Each U-Net unit is composed of two ResNet blocks (He et al., 2016), an attention block, and a convolutional up/downsampling block. We employ six U-Net units for both the downsampling and upsampling processes. In the downsampling units, the channel dimension is $64 \times j$, where $j$ represents the index of the layer ranging from 1 to 6. The upsampling units follow the reverse order. Each encoder module comprises a ResNet block and a convolutional downsampling block. For embedding conditioning, we employ ResNet blocks and transposed convolutions to upscale $\mathbf{z}$ to match the spatial dimension of the inputs of the initial four U-Net downsampling units. This allows us to perform conditioning by concatenating the output of the embedder with the input of the corresponding U-Net unit.

Figure 6 also describes our design choice of the model. We list the additional detailed specifications that we did not clarify in the main paper as follow:

- The hyper prior structure shares the same design as Minnen et al. (2018). The channel number of the hyper latent $\mathbf{y}$ is set as 256.

---

[1]https://github.com/Justin-Tan/high-fidelity-generative-compression
[2]https://github.com/Dezhao-Wang/Neural-Syntax-Code

- We use 3x3 convolution for most of the convolutional layers. The only exceptions are the 1st conv-layer of the first DU component and the 1st layer of the 1st ENC component, where we use 7x7 convolution for wider receptive field.
- $i_N^n$ is embedded by a linear layer, which expand the 1-dimensional scalar to the same channel size as the corresponding DU/UU units. We then add the expanded tensor to the intermediate ResBlock of each DU/UU unit.

## C   Model Efficiency

We provide information on the model parameter size of the proposed model and baselines, and the corresponding time cost of running a full forward pass in Table 1. We run benchmarking on a server with a RTX A6000. We decode 24 images from Kodak dataset and calculate the average neural decoding time, which does not include entropy-coding process, as all the model share a similar entropy model.

|  | CDC (1 step) | CDC (17 steps) | HiFiC | MS-hyper | DGML |
|---|---|---|---|---|---|
| Number of Parameters | $53.8M$ | $53.8M$ | $181.47M$ | $17.5M$ | $26.5M$ |
| Decoding Time (Seconds) | 0.015 | 1.04 | 0.0051 | 0.0011 | 0.0025 |

Table 1: Model and decoding time.

Our model exhibits superior memory efficiency compared to HiFiC. However, diffusion models suffer from slow decoding speed owing to their iterative denoising process. In the benchmark model utilized in our main paper, the decoding of an image takes approximately 1 second. Although this is slower than the baselines, it remains within an acceptable time range. Further optimization of the neural network module, such as the removal of the attention module, holds the potential to enhance efficiency even further.

## D   Additional explanation on experiment metrics

**FID**, as the most popular metric for evaluating the *realism* of images, measures the divergence (Fréchet Distance) between the statistical distributions of compressed image latent features and ground truth ones. The model extracts features from Inception network and calculates the latent features' corresponding mean and covariance. **LPIPS** measures the l2 distance between two latent embeddings from VGG-Net/AlexNet. Likewise, **PieAPP** provides a different measurement of perceptual score based on a model that is trained with the pairwise probabilities. **DISTS** measures the structural and textural similarities based on multiple layers of network feature maps and an algorithm inspired by SSIM. **CKDN** leverages a distillation method that can extract a knowledge distribution from reference images, which can help calculate the likelihood of the restored image under such distribution. Both **MUSIQ** and **DBCNN** are non-reference metrics, as they both use deep network models (transformer and CNN, respectively) that are pre-trained on labeled image data with Mean Opinion Score. For non-learned metrics (model-based methods), **FSIM** uses the phase congruency and the color gradient magnitude of two images to calculate the similarity. The **(CW/MS-)SSIM** family uses insights about human perception of contrasts to construct a similarity metric to mimic human perception better. **GMSD** evaluates the distance between image color gradient magnitudes. **NLPD** means normalized Laplacian pyramid distance, which derives from a simple model of the human visual system and is also sensitive to the contrast of the images. **VSI** reflects a quantitative measure of visual saliency that is widely studied by psychologists and neurobiologists. **MAD** implements a multi-stage algorithm also inspired by the human visual system for distortion score calculation.

# E    Supplemental Ablation Study

By manipulating the trade-off parameter $\rho$, our model can be trained to prioritize either perceptual quality or traditional distortion performance. In Figure 7, we present the rate-distortion curves for the COCO dataset. Throughout our study, we examine four distinct values of $\rho$ $(0, 0.32, 0.64, 0.9)$. The outcomes illustrate that higher values of $\rho$ result in improved perceptual quality but at the cost of worsened distortions in most scenarios. It is worth noting that values exceeding $\rho > 0.9$ are not viable, as perceptual quality ceases to exhibit noticeable improvements.

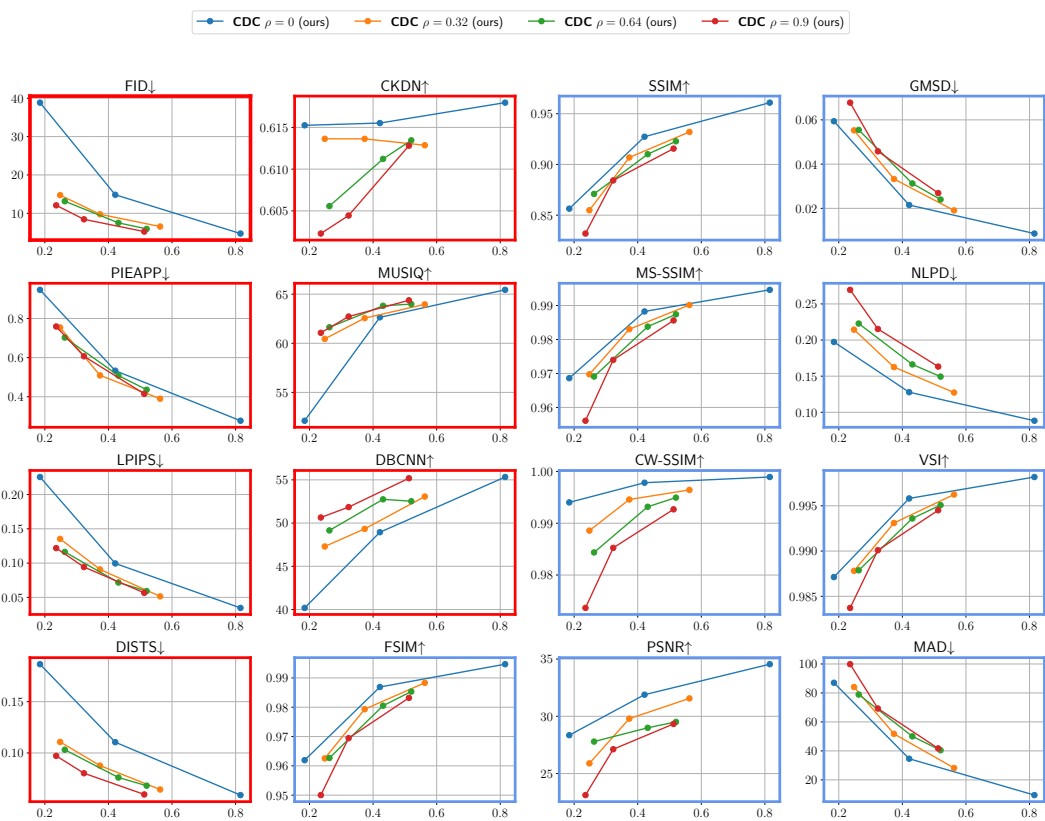

Figure 7: rate-distortion curves with different $\rho$ values

# F    Additional visualization of the compressed images and decoding variability visualization

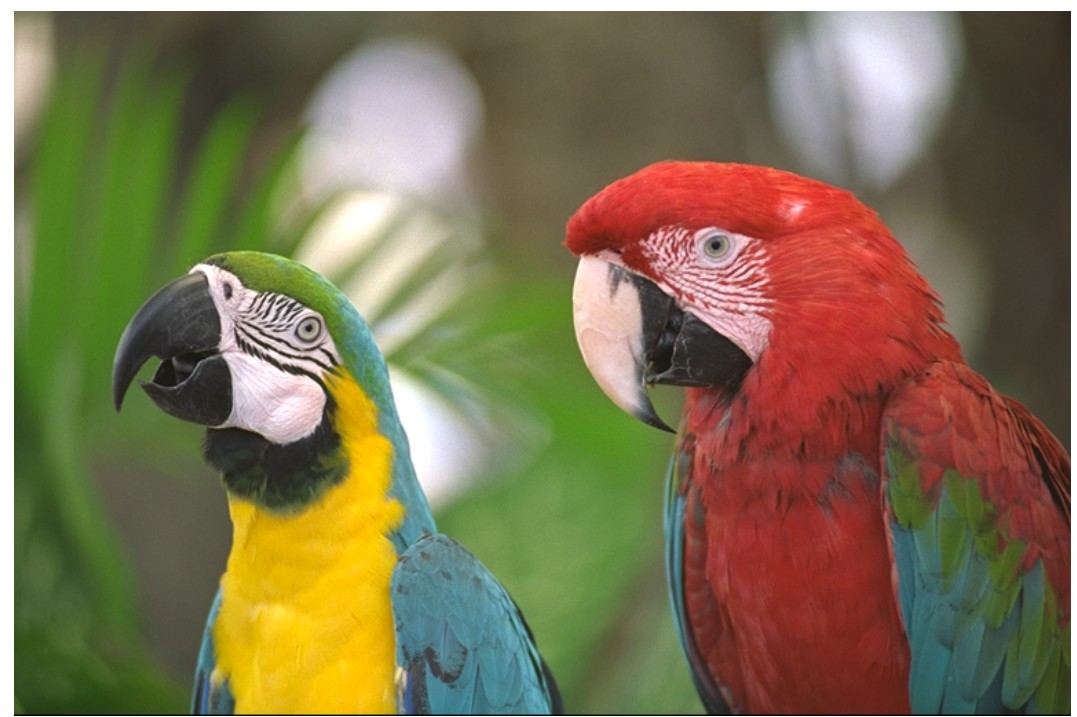

Figure 8: Ground Truth

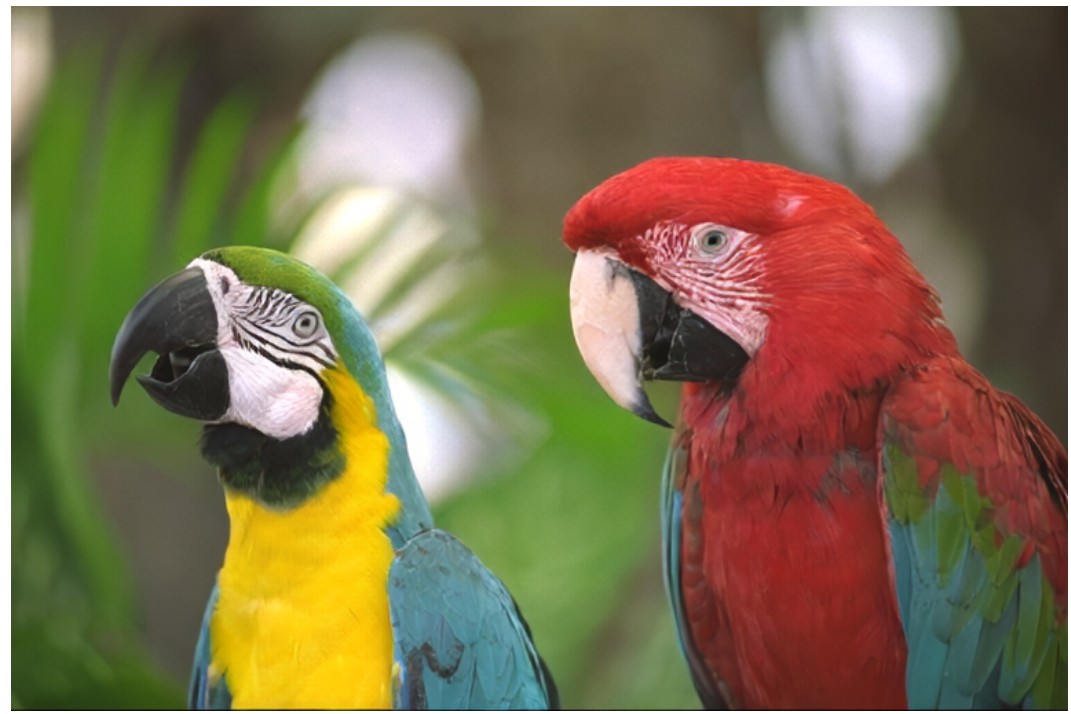

Figure 9: CDC $\mathcal{X}_\theta(\rho = 0.9)$, bpp=0.205

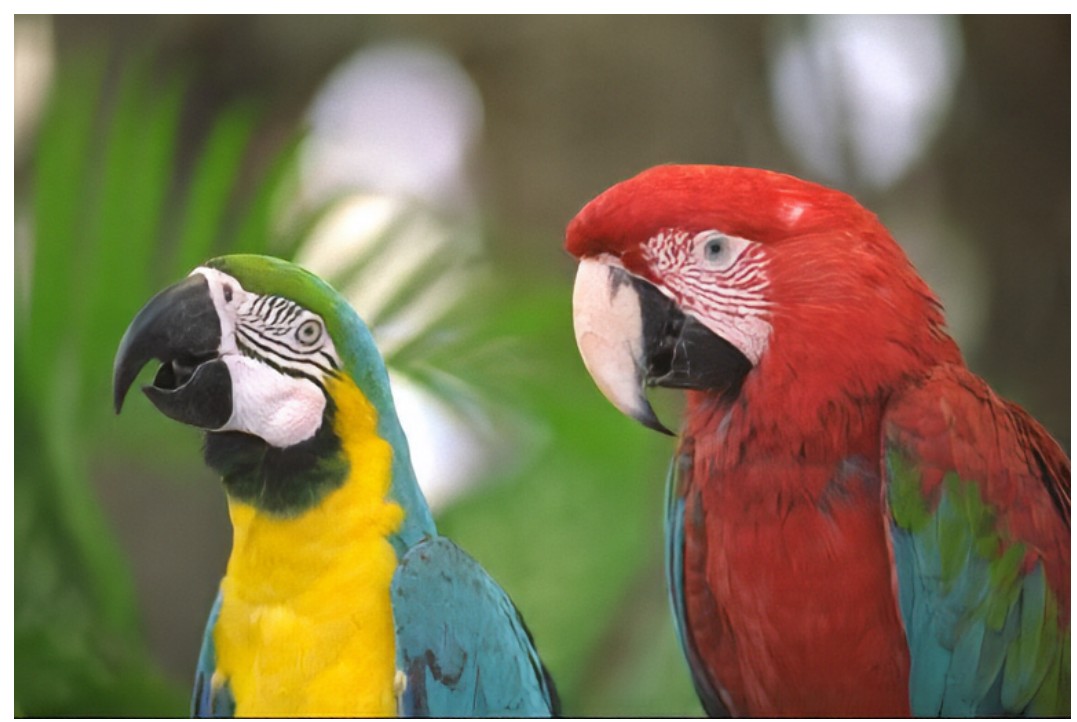

Figure 10: HiFiC bpp=0.207

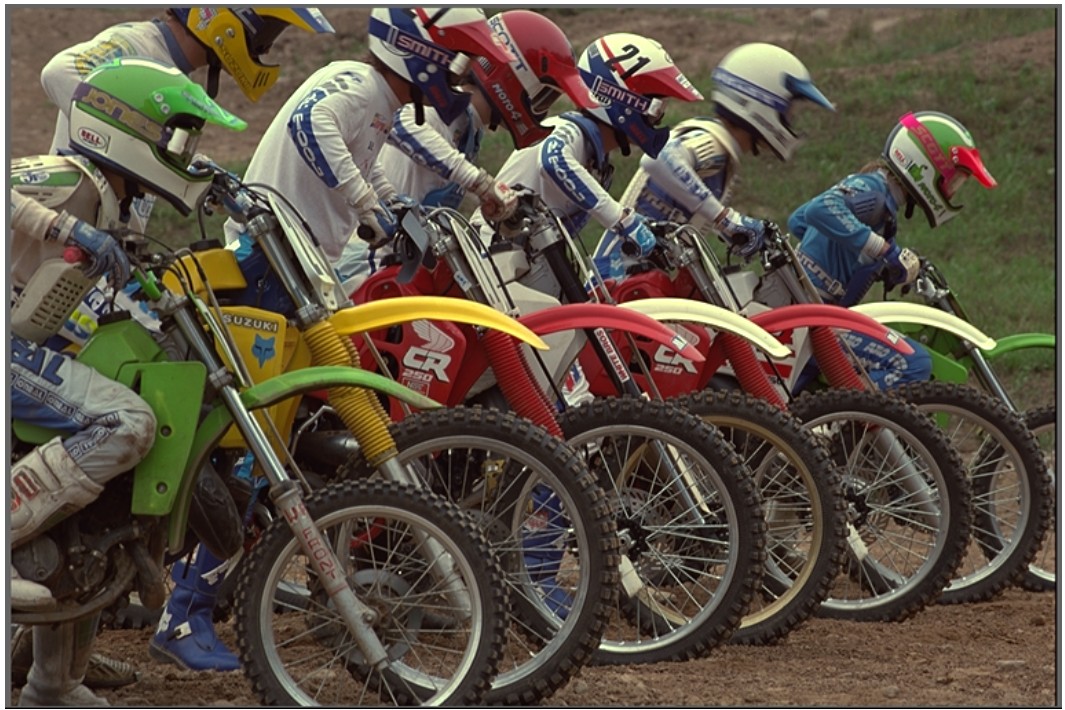

Figure 11: Ground Truth

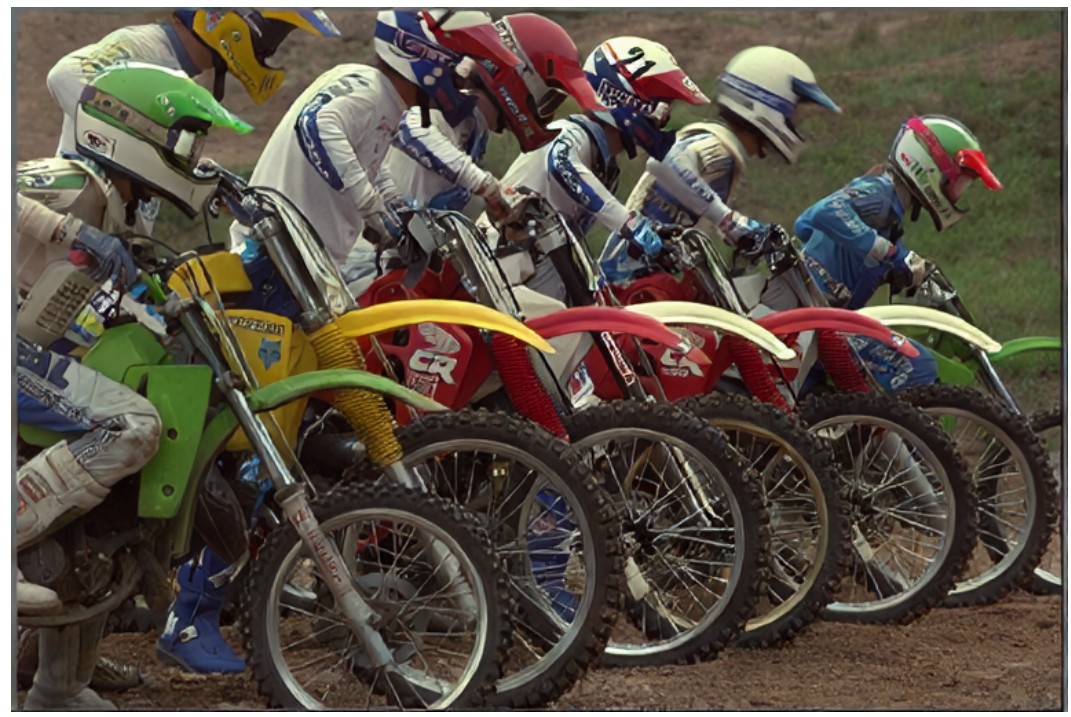

Figure 12: CDC $\mathcal{X}_\theta(\rho = 0.9)$, bpp=**0.398**

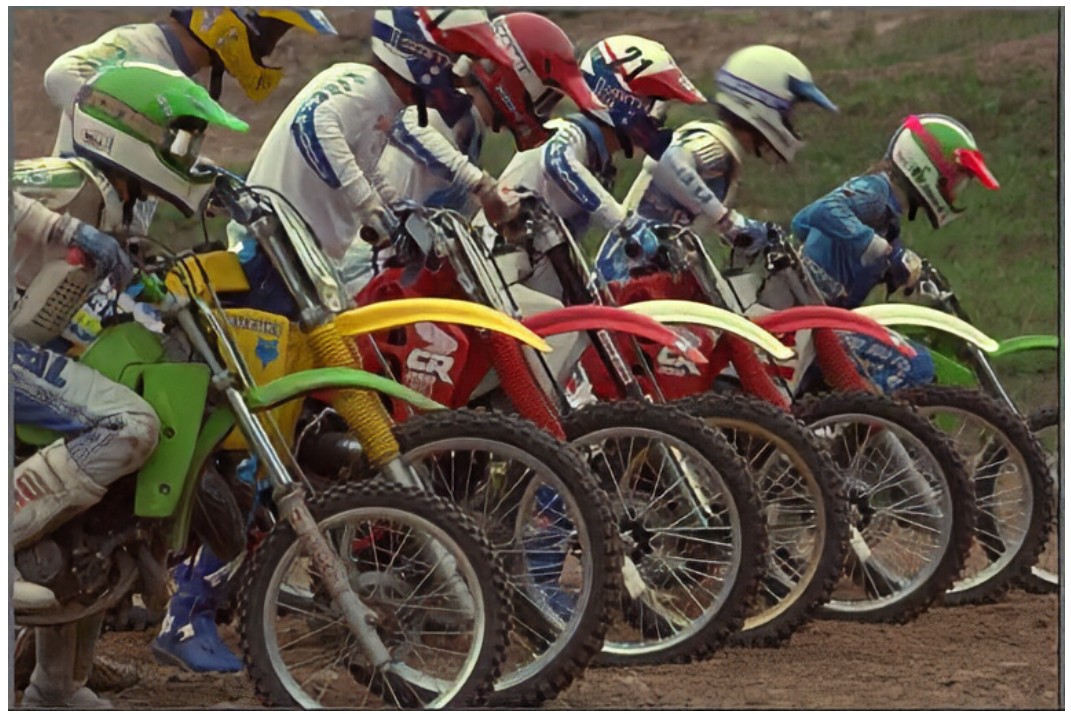

Figure 13: HiFiC bpp=0.456

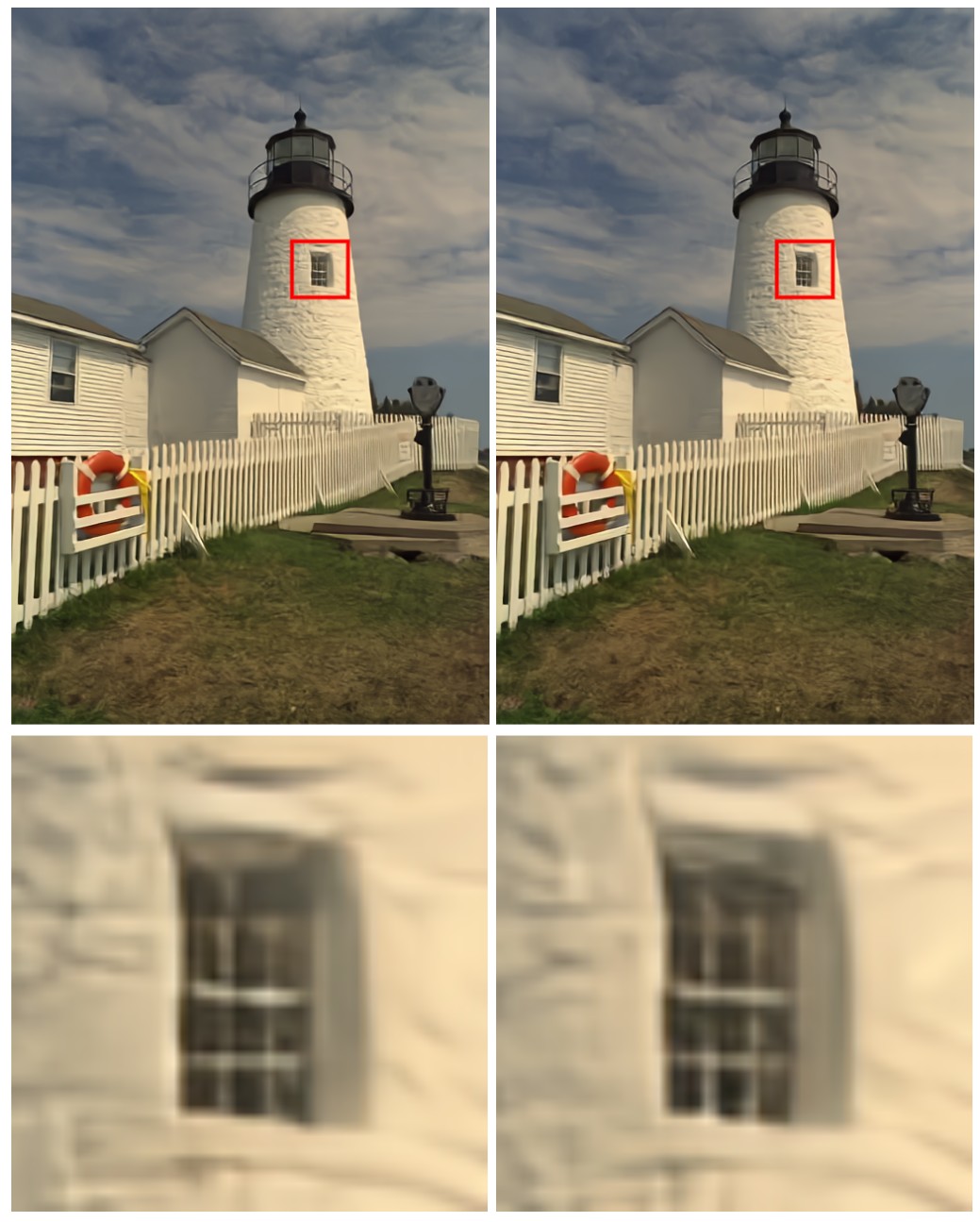

Figure 14: We stochastically decode the same latent variable $\mathbf{z}$ and $\gamma = 0.8$ but different random seed for $\mathbf{x}_N \sim \mathcal{N}(\mathbf{0}, \gamma^2\mathbf{I})$. Different random seeds may yield low-level textural distinction.

# G   Visualizations of the Decoding Process

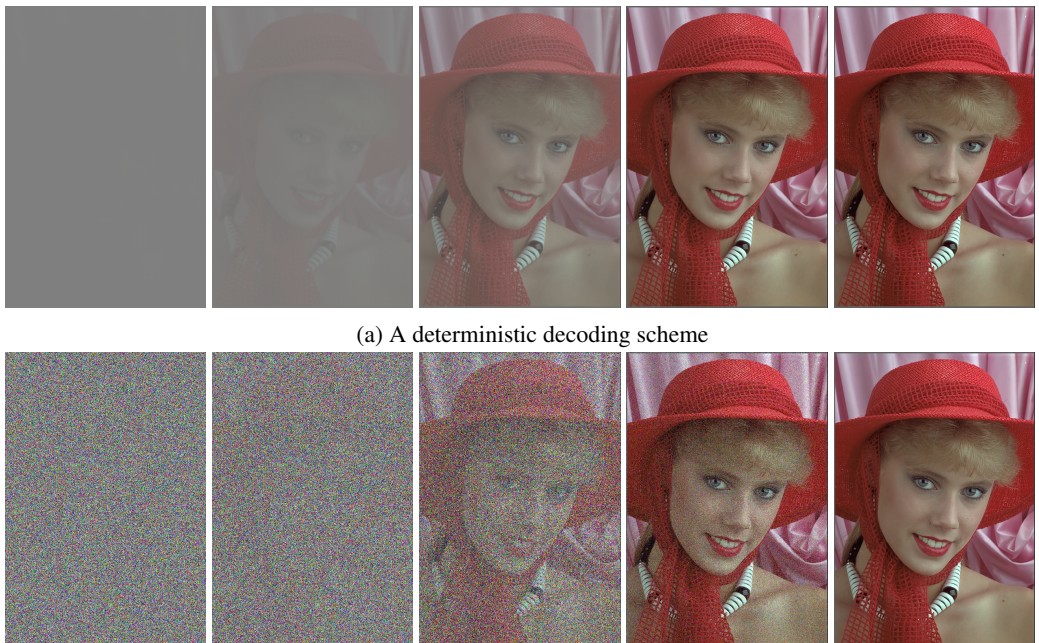

(a) A deterministic decoding scheme

(b) A stochastic decoding scheme

Figure 15: Comparing two decoding schemes for decoding the same image, this visualization illustrates the evolution of the texture variable $\mathbf{x}_n$ at five different time steps ($n = \{0\%N, 30\%N, 60\%N, 90\%N, 100\%N\}$) using a total of $N$ decoding steps. In the deterministic decoding scheme, the model transforms a grayscale image into the actual reconstruction. Conversely, in the stochastic decoding scheme, noise is dynamically "denoised" during the process, resulting in a reconstruction that incorporates high-frequency details.

# H    Additional Rate-Distortion(Perception) Results

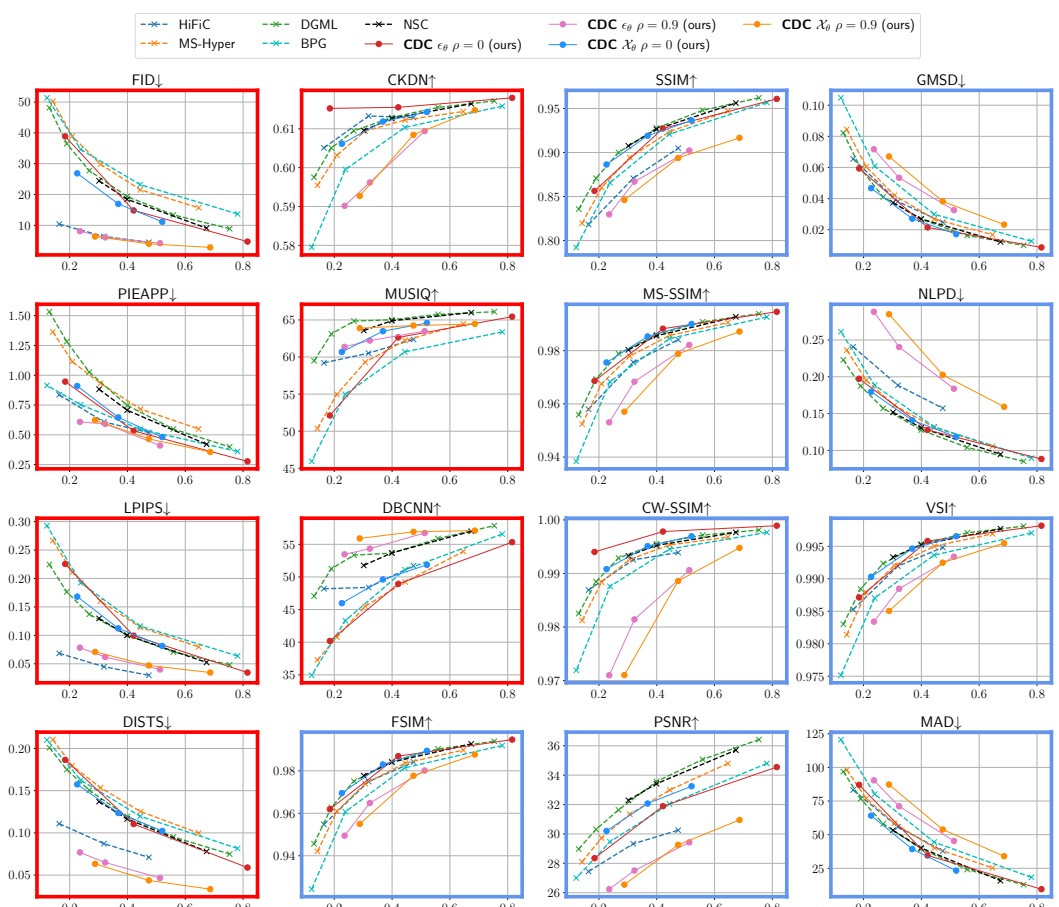

Figure 16: Rate-Distortion(Perception) for COCO dataset. We use 500 decoding steps for $\epsilon_\theta$ model.

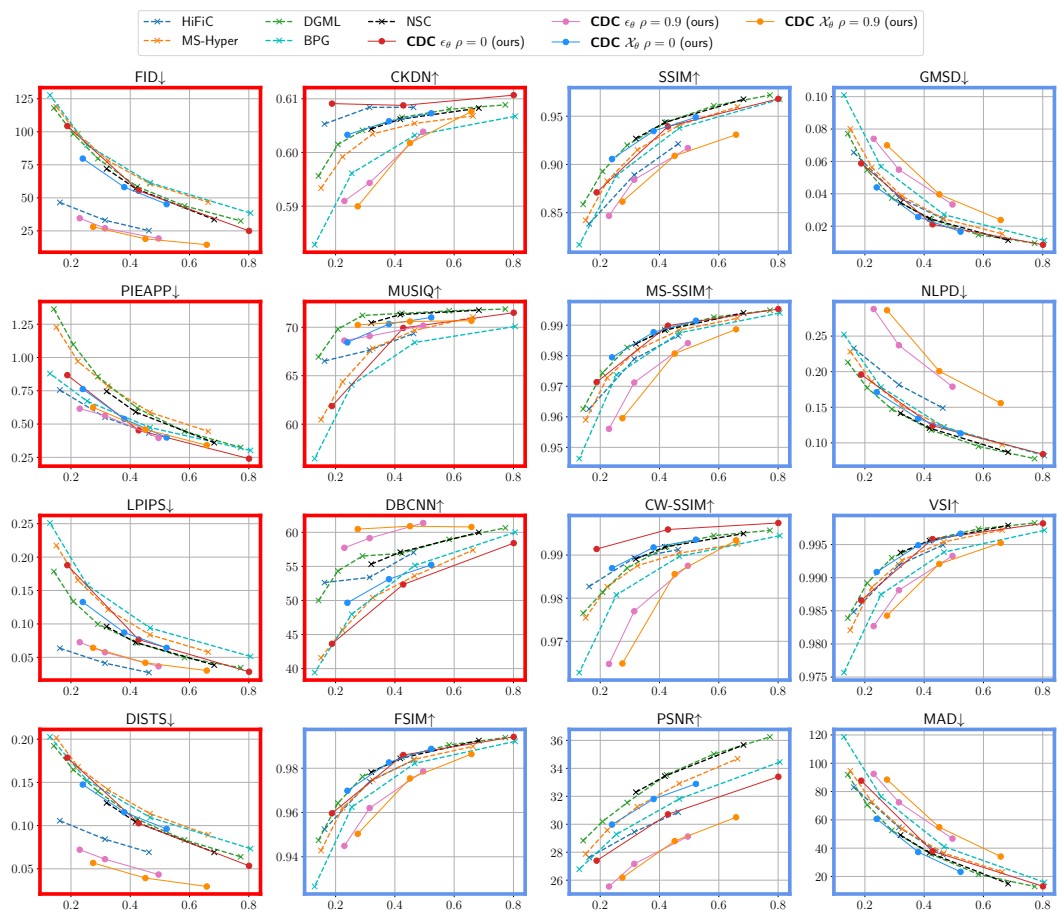

Figure 17: Rate-Distortion(Perception) for Tecnick dataset. We use 500 decoding steps for $\epsilon_\theta$ model.

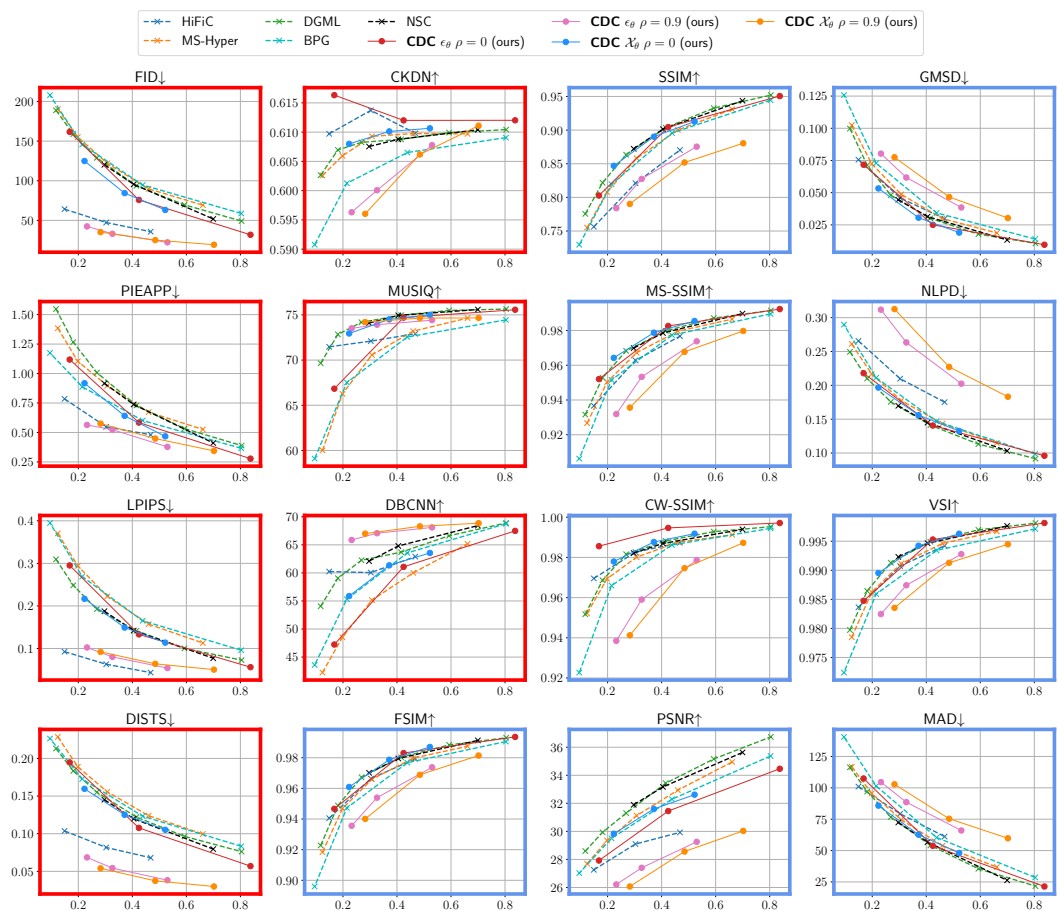

Figure 18: Rate-Distortion(Perception) for Kodak dataset. We use 500 decoding steps for $\epsilon_\theta$ model.

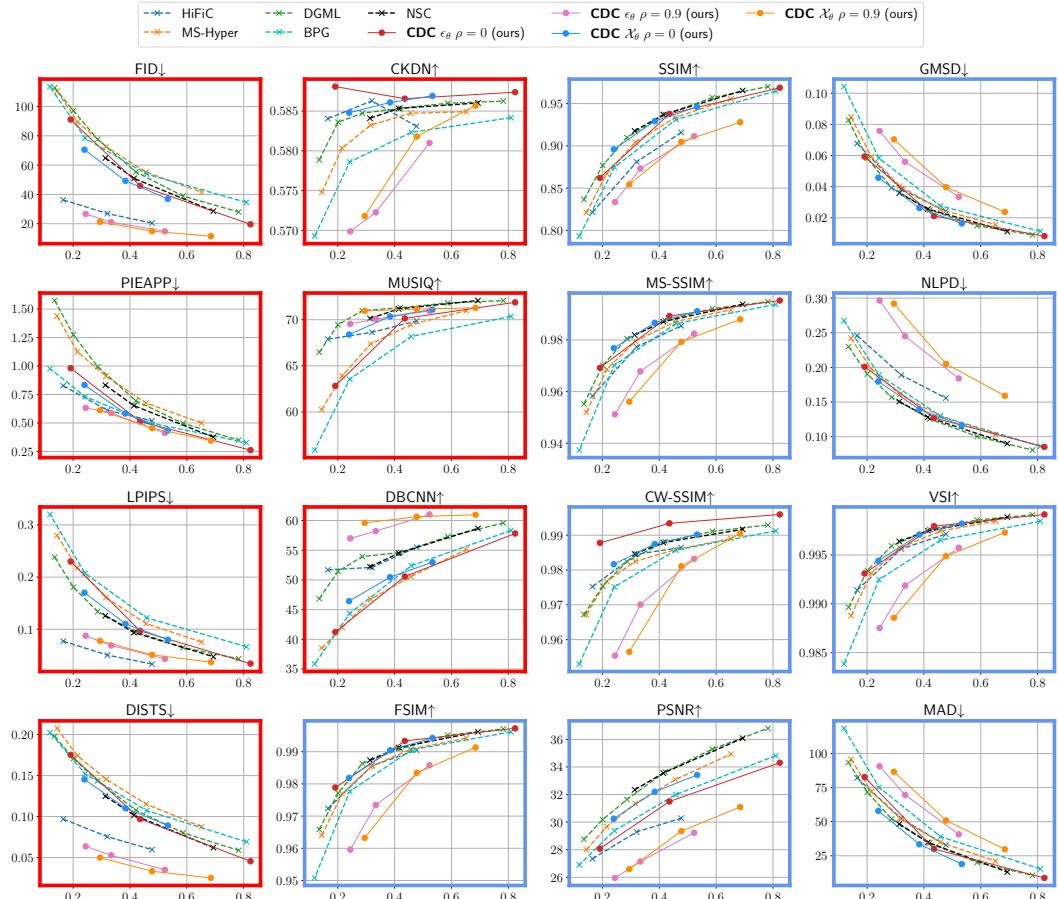

Figure 19: Rate-Distortion(Perception) for DIV2K dataset. We use 500 decoding steps for $\epsilon_\theta$ model.. The complete 16 metrics.

