# OpenReview forum: "Lossy Image Compression with Conditional Diffusion Models"
_NeurIPS.cc/2023/Conference — NeurIPS 2023 poster_

### Official Review · Reviewer_9a7p · 2023-07-04

**Soundness:** 3 good
**Presentation:** 2 fair
**Contribution:** 3 good
**Rating:** 6
**Confidence:** 3

**Summary:**

In this paper, the authors propose a novel architecture of lossy image compression using conditional diffusion models. They use an encoder to map images into contextual latent variables and train a conditional diffusion model under the guidance of these contextual latent variables as the decoder.  Moreover, they add perceptual loss into the loss of diffusion model, so this approach can steer the tradeoff between distortion, rate and perceptual quality by changing the parameter of its loss and diffusion model. Finally, they conduct sufficient experiments to explore how the parameters influence the final performance and to compare with other approaches.

**Strengths:**

1. The architecture based on the conditional diffusion model is interesting and creative, and as is shown from the experiments in 3.2, this architecture can also finish the reversed progress of the conditional diffusion model in a handful of steps.
2. Sufficient experiments have been conducted, including comparison with other methods in sixteen metrics, and exploration of how parameters like ρ, γ, and decoding steps influence the final performance.
2. Overall the results prove that this method can steer the tradeoff between distortion, rate and perceptual quality, and the final performance with specific parameters in the paper is also quite competitive in different metrics.

**Weaknesses:**

1. Some implementation details are not illustrated clearly, including how to get the "17 steps" of the reversed progress in 4.1 which may be helpful to understand the schedule of the reversed progress, the cost of encoding and decoding time, and I think a picture including the image of every single step in 17 steps when predicting x0, may help to prove that the architecture can reconstruct the "texture" structure of the image in a handful of steps.

2. More qualitative and quantitative results may be helpful to prove the performance and practicability of the method. including images with different resolutions and different styles of images.

2. I think, when testing, the parameters of the diffusion model should be given more specifically, including the total number of steps, the true number of steps after resampling, and the type of schedule(cosine or linear), because these parameters can influence the decoding steps metioned in the paper.

3. In formula (7), I think it can only represent the situation of γ=0, that is the DDIM, but when γ>0, I think there should be an extra random noise according to the formula(predicting noise is changed to predicting x0)

   $$
   \boldsymbol{x}_{t-1}=\sqrt{\alpha_{t-1}} \underbrace{\left(\frac{\boldsymbol{x}_t-\sqrt{1-\alpha_t} \epsilon_\theta^{(t)}\left(\boldsymbol{x}_t\right)}{\sqrt{\alpha_t}}\right)}_{\text {"predicted } \boldsymbol{x}_0 "}+\underbrace{\sqrt{1-\alpha_{t-1}-\sigma_t^2} \cdot \epsilon_\theta^{(t)}\left(\boldsymbol{x}_t\right)}_{\text {"direction pointing to } \boldsymbol{x}_t \text { " }}+\underbrace{\sigma_t \epsilon_t}_{\text {random noise }}
   $$

**Questions:**

1. In section 4, I wonder why the images of the Kodak dataset are not resized while the others are resized, and how the diffusion model trained on an image size of 256×256 can be applied to images of different sizes when testing.

**Limitations:**

Yes, the authors have considered the limitations and societal impact of neural compression, particularly when prioritizing perceptual quality, is the risk of misrepresenting data in the low bitrate regime.

---

> ### Author Rebuttal · Authors · 2023-08-09
>
> > Some implementation details are not illustrated clearly
>
> We have supplied the decoding time of the model and architecture details in Appendices A and C. Additionally, Appendix G contained visualizations of image transitions during the decoding process. We chose 17 as the number of steps to strike a balance between achieving satisfactory perceptual quality and maintaining a reasonable decoding time.
>
> > More qualitative and quantitative results.
>
> We had more visualizations in Appendix F. The rate-(distortion/perception) curves for all of our datasets were available in Appendix H. Regarding your valuable suggestion of plotting image x_n over iterations: note we already included exactly this in Appendix G Figure 15.
>
> >  the parameters of the diffusion model should be given more specifically
>
> In Appendix B, we have included comprehensive images and descriptions of our model's architecture. We employ a linear noise schedule for the diffusion process, and we apologize for not mentioning this in the paper and we also found that there are not big performance gaps between cosine schedules and the linear ones. We are not sure what you mean by “true number of steps after resampling”, could you elaborate on this? As stated in our experiment section, we used 20,000 steps for training the epsilon-prediction model and 8,000 steps for the x-prediction model. The 17 steps refer to the number of decoding steps utilized for decoding during the main evaluation. We will refer to the appendix in the main paper more explicitly.
>
> > In formula (7), I think it can only represent the situation of γ=0
>
> There may be a misunderstanding of the source of stochasticity in DDIM. DDIM converts an original stochastic differential equation (DDPM) into a deterministic ODE; the only source of noise is in the initial state of $x_N$. All our algorithms use this DDIM scheme, which was shown to provide better sample qualities than following the stochastic differential equation especially under small sampling steps (see [Song et al. 2020](https://arxiv.org/pdf/2010.02502.pdf)). Note that when we say deterministic/stochastic in our model, we only refer to whether $x_N$ is fixed to 0 or stochastically sampled from $N(0, \gamma^2I)$. The $\gamma$ term solely controls the variance of the initial noise $x_N$. The noise term in your formula is weighted by $\sigma_n$, which is fixed to zero for DDIM.
>
> >  I wonder why the images of the Kodak dataset are not resized while the others are resized. how the diffusion model trained on an image size of 256×256 can be applied to images of different sizes when testing.
>
> The Kodak dataset is a widely-used image compression benchmark dataset containing high-quality images. For the other datasets, we resize the images to eliminate compression artifacts introduced by their original codecs, such as JPEG. This is common in the literature since otherwise JPEG compression would not incur additional compression losses and have an unrealistic advantage. It is important to note that we employed a maximum of 6 downsample and 6 upsample layers with a scale factor of 2, which implies that the width/height of images would ideally be a factor of 64. Lastly, our model is fully convolutional, enabling it to handle various image sizes.

---

> > ### Comment · Reviewer_9a7p · 2023-08-18
> >
> > Thank you for your response and most of my concerns are responded. I decide to raise my rating.

---

### Official Review · Reviewer_ys1i · 2023-07-06

**Soundness:** 4 excellent
**Presentation:** 4 excellent
**Contribution:** 3 good
**Rating:** 6
**Confidence:** 3

**Summary:**

This paper presents a method of using conditional diffusion as a stochastic decoder for neural image compression. The approach learns a discrete latent representation of an image and conditions the diffusion process on this representation. Compression is achieved by learning a distribution over the latent representation and running an entropy coder/decoder as is commonly done in other neural image compression models.

The main benefit of this approach is that diffusion (empirically) does a better job that perceptual metrics (like LPIPS) or adversarial losses in boosting the "realism" or "perceptual quality" of reconstructed images. The paper shows that with multiple metrics (Fig. 2) and images (Fig. 3).

The paper also shows that relatively few diffusion steps are required to generate high-quality reconstructions (Fig. 4) due to the conditioning and the use of X-prediction instead of predicting epsilon (the noise for step t). I didn't see runtime numbers in the paper, but the reduction of diffusion steps should lead to "moderate" decode times (contrast with early diffusion methods that required hundreds or thousands of steps).

**Strengths:**

The main strength of the paper is the effective use of a diffusion-based decoder for neural image compression that leads to high subjective/perceptual visual quality according to both quantitative metrics and the example reconstructions in the paper. In particular, the results appear better than those from similar, GAN-based models (like HiFiC), and the ability to adjust the stochasticity at inference time (by adjusting gamma, which controls the amount of noise used by the reverse-diffusion process) is quite interesting and potentially helpful reducing concerns about hallucinated details.

The paper is also very well-written and easy to follow, and diffusion-based stochastic decoders are of high interest to the neural image/video compression community.

**Weaknesses:**

The main weakness of the paper is low novelty since diffusion for image generation is well-known at this point so its use in a codec is a fairly obvious application (though it's not technically easy to achieve good results).

The authors should cite this paper on the same topic, though as far as I know it has not been published in a peer-reviewed, archival venue so it doesn't block publication of submission 3900:

Hoogeboom 2023. High-Fidelity Image Compression with Score-based Generative Models
https://arxiv.org/abs/2305.18231

The authors should also compare, if possible, against more recent GAN-based models that outperform HiFiC:

PO-ELIC: Perception-Oriented Efficient Learned Image Coding
He 2022. https://arxiv.org/abs/2205.14501

Multi-Realism Image Compression with a Conditional Generator
Agustsson 2023. https://arxiv.org/abs/2212.13824


The inclusion of runtime numbers and a rater study would also strengthen the paper, but I don't think a rater study should be required for publication for every compression paper.



**Questions:**

If I understand the formulation properly, rho controls the weight on LPIPS loss used during training (Eq. 9). In Fig. 2, I'm surprised there's such a large gap between the "CDC \rho=0" (blue) and "CDC \rho=09" (orange) curves for FID. It makes sense that there's a large gap for LPIPS and DISTS since high values of rho mean you're using the test metric (or close to it in the case of DISTS) as part of the training process (to be clear: training with LPIPS is common so I'm not implying this is a problem). But I would have expected diffusion alone to improve FID scores.

So is LPIPS doing the heavy lifting here, at least relative to the metrics? Is there a better metric for measuring the benefit of using diffusion vs. a GAN (plus LPIPS) loss as in HiFiC?


**Limitations:**

adequately addressed

---

> ### Author Rebuttal · Authors · 2023-08-09
>
> > not novel enough
>
> Please see general response 1
>
> > The authors should cite this paper on the same topic
>
> Thank you for pointing this out; we also noticed these important related work and will cite and discuss it in the final paper version. Note that some works are concurrent and appeared after this submission.
>
> > The inclusion of runtime numbers and a rater study
>
> We reported runtime in Appendix C. We agree that a rater study could be an interesting option for image quality evaluation. While user studies are clearly very nice and valuable, we need to consider our available resources of funds to include such a study in the camera-ready version.
>
> > I would have expected diffusion alone to improve FID scores. LPIPS is doing a heavy lift.
>
> We agree that this is a surprising outcome of our study, but was rigorously tested across 16 metrics. Please refer to general response 2 for details. Since the most important perceptual metrics are still subject to an ongoing debate, it's hard to say if there is a better metric.

---

### Official Review · Reviewer_wzSD · 2023-07-07

**Soundness:** 2 fair
**Presentation:** 2 fair
**Contribution:** 3 good
**Rating:** 5
**Confidence:** 4

**Summary:**

This paper proposes a diffusion-based image codec. It has two major components: a VAE-based encoder and a diffusion-based decoder. The VAE-based encoder generates the image latents, which serve as the conditioning signal for the diffusion-based image reconstruction. Both components are trained end-to-end with an objective function that involves a rate and a distortion estimate. The rate estimate is from the hyerprior model while the distortion estimate is a weighted sum of the perceptual loss and the mean squared error between the denoised image and the original image.

**Strengths:**

(1) The idea of combining a VAE-based encoder and a diffusion-based decoder for learned image compression in an end-to-end trainable framework is novel and new.

(2) Experimental results look interesting and promising.

**Weaknesses:**

(1) It is rarely seen that a diffusion model needs to be trained with a perceptual loss in order to achieve better subjective quality. From Fig. 2, without the perceptual loss (i.e. \rho=0), the proposed method does not really stand out among the competing methods in terms of perceptual metrics. More insights need to be provided.

(2) It is unclear whether one could use more decoding steps without the perceptual loss to achieve the same perceptual quality as with the perceptual loss with fewer decoding steps. This may justify the use of the perceptual loss from the perspective of the decoding complexity.

(3) When examining carefully the training objective, I notice that L_upper is weighted additionally by the noise schedule coefficient \alpha_n whereas the perceptual loss is not. Does this suggest that the perceptual loss may dominate the distortion estimation? I wonder what would happen if \rho is increased beyond 0.9.

(4) Surprisingly, in this work, the x-prediction (17 steps) is able to achieve better performance than the \epsilon prediction (500 steps) with fewer decoding steps. This is a bit counterintuitive and is different from how the diffusion model is typically trained. Is there any explanation behind this observation? A more thorough comparison between these two design choices should be provided to justify the use of the x-prediction.

(5) In Figure 4, the distortion curves are visualized with r=0 whereas the perceptual curves are with r=1. How about the other way around, r=1 for distortion and r=0 for perceptual?

(6) In Appendix B, is there any reason for adding the z embedding to only the first four downsampling blocks instead of all the blocks?

**Questions:**

Please see my points in the Weaknesses box.

**Limitations:**

The authors may like to comment on the reproducibility of the proposed method. For example, can experts in this field re-produce the results easily by following the training procedure given in this paper? Or the training may involve lots of "know-hows".

---

> ### Author Rebuttal · Authors · 2023-08-09
>
> > It is rarely seen that a diffusion model needs to be trained with a perceptual loss.. More insights need to be provided
>
> Please see our general response 2
>
> > It is unclear whether one could use more decoding steps without the perceptual loss to achieve the same perceptual quality as with the perceptual loss with fewer decoding steps
>
> More denoising steps can improve the perceptual quality. As we showed in Figure 4, by increasing the number of denoising steps exponentially, the perceptual quality improves but saturates. Unfortunately, this improvement still cannot fill in the gap between models with and without perceptual loss.
>
> >  Does this suggest that the perceptual loss may dominate the distortion estimation?.. I wonder what would happen if $\rho$ is increased beyond 0.9
>
> This is a good question. We add a new plot in our general response Figure 1, which suggests that this does not seem to be the case. If the perceptual loss were to significantly dominate the loss function and overshadow the stochastic score matching loss, a single x-prediction step should achieve similar reconstruction quality as multiple reconstruction steps, as $\mathcal{X_\theta}(x_n, z, n)$ may approximate $\mathcal{X_\theta}(z)$. However, our approach's performance markedly improves with multiple diffusion steps (1 vs 17). This evidence indicates that both LPIPS and diffusion losses play crucial roles in our methodology.
>
> Moreover, it is important to note that the weighting term $\frac{\alpha_n}{1-\alpha_n}$ dynamically changes according to the subsample index n, potentially becoming very large when $\alpha_n$ approaches 1, or very small as $\alpha_n$ approaches 0. We also discussed the impact of the $\rho$ value in Appendix E. While larger $\rho$ values do result in higher perceptual quality, it will also saturate. $\rho>0.9$ will not be able to provide prominent perceptual quality improvements.
>
> >  the x-prediction (17 steps) is able to achieve better performance than the \epsilon prediction (500 steps) with fewer decoding steps… counterintuitive
>
> Please see our general response 3.
>
> > the distortion curves are visualized with r=0 whereas the perceptual curves are with r=1. How about the other way around
>
> We actually did these experiments as well and decided at the last minute to simplify the figure and not show them for simplicity. We provide the remaining figures for completeness in general rebuttal Figure 2.
>
> > is there any reason for adding the z embedding to only the first four downsampling blocks
>
> This is due to the U-Net architecture our diffusion model uses. Our neural encoder (ENC) consists of only four downsampling layers; therefore, only the first four U-Net layers correspond to the spatial dimensions of the outputs from the four upsampling EMD layers. This is the rationale behind concatenating only four layers instead of six.
>
> > Reproducibility
>
> See our general response 4

---

### Official Review · Reviewer_QC9o · 2023-07-08

**Soundness:** 3 good
**Presentation:** 3 good
**Contribution:** 3 good
**Rating:** 5
**Confidence:** 4

**Summary:**

In this paper, a conditioned diffusion model is proposed to improve the perceptual quality of learned image compression. Different from widely used VAE-like nonlinear transform coding frameworks, the proposed model adopts a diffusion process to synthesize most of the texture, conditioned on a learned hyperprior variable z. To further improve the perceptual quality, a mixed objective is proposed, combining the ELBO target of the diffusion model with a perceptual loss term. The proposed approaches show reasonable performance on most perceptual metrics, surpassing baselines like HiFiC.

**Strengths:**

This is overall an interesting paper. The authors intend to borrow the very success of diffusion models to the LIC approach, to promote the perceptual quality due to the improved generation ability. The proposed approach of using DDPM/DDIM as a conditioned texture synthesizer is intuitive and easy to follow. The presented results are overall impressive and encouraging. They present a new diagram for this line of research, which tend to inspire many future studies.

**Weaknesses:**

1. The approach seems somewhat empirical. We can explain many of L1-LPIPS-GAN joint losses i.e. the one adopted by HiFiC as it corresponds to an implicit likelihood distribution (discussed in Ball'e 2018 hyperprior paper, typically an energy-based model) or some divergence measure (see f-gan). Let I cannot connect a similar theoretical analysis with the objective described in eq.8 & eq.9. Though this perceptual loss term is intuitive, I suggest the authors provide some more theoretical analysis to convince.
2. I wonder why the LPIPS performance of CDC is worse than HiFiC, especially since the model is directly supervised under an LPIPS perceptual loss. A better FID (generation quality) with worse LPIPS (perceptual fidelity) is somewhat suspicious. Can we explain this because the diffusion models generate more 'hallucination' texture seems true but with lower fidelity to the ground truth? This may be true according to Figure 3(b), some artifact black-white textures near the eye not exiting in GT are generated in CDC, from my perspective.
3. It may be unfair to compare the proposed model directly with HiFiC, as the network architecture of HiFiC is relatively lighter with less parameter volume than the UNet. The authors should report the model size and discuss the influence of improved model capacity. It may be better to discuss some larger LIC baselines eg PO-ELIC or evaluate a smaller UNet synthesizer.

**Questions:**

See above.

**Limitations:**

The authors have addressed the limitation.

---

> ### Author Rebuttal · Authors · 2023-08-09
>
> > The approach seems somewhat empirical... provide some more theoretical analysis to convince.
>
> Neural image compression is largely an empirical field, so our paper does not stand out in this regard. Yet, we made sure our optimization objective follows from a rigorous lower bound on a latent variable model’s intractable marginal likelihood. This is outlined in the text and equations following line 158. We also contrast the capabilities of our approach to HiFiC, which optimizes a GAN-like loss instead.
>
> > I wonder why the LPIPS performance of CDC is worse than HiFiC…
>
> We note that both our method and HiFiC employ LPIPS as an auxiliary loss term, suggesting that HiFiC might also exhibit strong performance in LPIPS measurements. However, our approach differs from GANs, which learn a more rigid mapping from the latent variable $z$ to the decoded images. In contrast, our diffusion models incorporate a noise term $x_N$, which introduces perturbations. This noise-induced variability may result in suboptimal LPIPS outcomes but yields better FID scores, highlighting a key distinction between our method and GAN-based approaches. Thus, our diffusion models offer unique benefits in terms of image quality and generative capabilities.
>
> > Can we explain this (follow the question above) because the diffusion models generate more 'hallucination' texture that seems true but with lower fidelity to the ground truth?
>
> Follow our answer above, yes. While the noise perturbation reduce distortion metrics including the “perceptual distortion” LPIPS, it yields better FID scores. And it indeed generates hallucination textures as you can see. We also discussed this point in the limitation section.
>
> > It may be unfair to compare the proposed model directly with HiFiC, as the network architecture of HiFiC is relatively lighter with less parameter volume than the UNet
>
> We have included essential information regarding the architecture and model size comparison between our model and HiFiC in Appendix A&C. Note that our approach utilizes a lightweight UNet, with the largest conv-layer containing only 384 channels. This results in 53.8 million parameters, versus HiFiC having 181.5 million parameters. The primary challenge for diffusion models lies in the iterative decoding process, which may result in slower decoding times. However, as we pointed out above, we think that the exploration of diffusion-based approaches should not be prematurely discarded on this basis as long as the decoding times are not completely off-the-charts. We consider one order of magnitude slower decoding times than traditional methods as an accomplishment and a reasonable basis for further development. We will defer the exploration of even faster decoding methods to future studies.

---

> > ### Comment · Reviewer_QC9o · 2023-08-21
> >
> > I appreciate the authors for their response to my concerns. While the rebuttal has addressed some of my points, I still have reservations regarding the FID/LPIPS gap between the proposed approach and existing VAE-like approaches such as HiFiC. When it comes to image generation tasks without ground truth, it becomes challenging to perform case-wise evaluations. In such cases, FID serves as a primary consideration since it provides a distribution-wise measurement, assessing whether the sampled textures align with the in-distribution data. This is why FID is typically calculated directly on a training set when evaluating generation tasks, as we expect the samples to adhere to the observed empirical data distribution.
> >
> > However, image compression is a task that has case-wise ground truth. FID is a coarser evaluation for this type of task. Since comparing FID between a pair of input/output images is meaningless, we cannot conduct case study to determine whether the quality of the compressed image is good or identify the specific shortcomings of an approach. It is unclear whether a good FID score on a decoded set indicates good consistency to inputs or an overemphasis on FID's preferences (ie the hallucination artifacts). On the other hand, LPIPS provides a finer-grained evaluation for paired assessments and has been shown to be highly correlated with human perception. An approach with a high LPIPS score is less likely to exhibit fake artifacts. Therefore, I still consider LPIPS as a primary reference for judging an image compression method.
> >
> > I acknowledge the value of exploring diffusion-based approaches for learned image compression and appreciate the authors' efforts in this regard. However, it is crucial to include both theoretical and experimental discussions since NIC is an empirical approach built on a solid foundation established by Ballé et al papers and VAE/ELBO theories. Given that this earlier exploration combining diffusion models with NIC challenges the existing framework and has the potential to significantly influence future studies, I maintain my rating score above the threshold but remain cautious.
> >
> > Overall, I appreciate the authors' response and encourage them to provide further clarification and evidence to support their proposed approach's superiority over existing VAE-like approaches.

---

> > > ### Author Response · Authors · 2023-08-21
> > > **About Metrics and HiFiC baseline**
> > >
> > > Thank you for taking the time to share your thoughts, but we still need to clarify some key points.
> > >
> > > 1. We would like to emphasize a key observation from the HiFiC paper's conclusion: "**By comparing our metrics with the outcome of the user study, we showed that no existing metric is perfect for ordering arbitrary models right now, but that using FID and KID can be a valuable tool in exploring architectures and other design choices.**" This assertion substantially reinforces the relevance of the evaluation metrics we employed.
> > >
> > > 2. Moreover, the aforementioned point reinforces our contention that relying solely on LPIPS for assessing perceptual image quality can introduce bias. While our LPIPS scores may appear less favorable than those of HiFiC, our superior performance across DIST, PieAPP, and FID metrics underscores our approach's broader effectiveness.
> > >
> > > 3. When calculating FID, it's crucial to recognize that we're essentially measuring the divergence between the **compressed test data distribution** and its corresponding **uncompressed test data distribution**. This underscores that the data remain paired at the distribution level, without resorting to the training data as a reference distribution.
> > >
> > > 4. Lastly, we must acknowledge that HiFiC, being a GAN-VAE hybrid method, is not exempt from the challenge of "fake artifacts," an issue they also candidly addressed in their Broader Impact section. As we mentioned in the paper, this rate-distortion-perception theory was sufficiently discussed by [Blau et al. 2019](https://arxiv.org/pdf/1901.07821.pdf)
> > >
> > > Once again, we appreciate your engagement and feedback, which allow for a more nuanced discussion of the intricacies involved.

---

### Official Review · Reviewer_kiiX · 2023-07-13

**Soundness:** 3 good
**Presentation:** 4 excellent
**Contribution:** 3 good
**Rating:** 5
**Confidence:** 5

**Summary:**

This paper proposes an end-to-end learned image compression method based on diffusion model, where it consists of a VAE for latent feature encoding and a conditional diffusion decoder. Concerning theoretical parts, the authors also derive the variational upper bound to the diffusion model’s negative data log likelihood. Apart from that, by training the diffusion model with X-parameterization, the model can enable high-quality reconstructions in only a handful of decoding steps. Experimental results on 4 test datasets with 16 different metrics show that the proposed method can achieve comparable results compared to existing methods.

**Strengths:**

1. The paper is well and clearly written. The method part also includes some theoretical derivations.
2. The method achieve sgood performance with extensive and complete evaluations on 4 different datasets.
3. The introduction of X-parameterization significantly reduces the long decoding time, which makes the proposed diffusion-based method promising for applications.

**Weaknesses:**

1. The method itself is reasonable but without a clear motivation. It would be nice if the authors could spend some words on discussing some key advantages of using diffusion model for generative-based LIC? E.g., it would be nice if there is a significant improvement in terms of perceptual quality compared to VAE-based generative LIC, or that diffusion-based LIC can have some other benefits that VAE-based cannot have.
2. Although with X-parameterization, the runtime is improved to be within a handful number of decoding steps, still as shown in the supplementary materials, the decoding time is still significantly longer than existing LIC methods.
3. It is also welcome to have as many as evaluation (e.g., 16 metrics) is good, however, without explanations or analysis, it can be hard for readers who are not familiar with some of the metrics to qualitatively interpret the results. Compared to more metrics, personally, i prefer reporting comparison results with more publicly available methods on only 2-3 representative or well-known evolution metrics in the main paper while putting the others in the supplementary materials.


**Questions:**

1. I am confused about the theoretical analysis or rationale behind why using the X-parameterization rather than the $\sigma$ parameterization can significantly reduce the number of decoding steps. I think this should be addressed in the rebuttal and also in the paper.
2. I am also wonder if the authors are planing to release their code?

---

> ### Author Rebuttal · Authors · 2023-08-09
>
> > The method itself is reasonable but without a clear motivation.. What’s the key advantages
>
> Please see our general response 1 above.
>
> > Although with X-parameterization..decoding time is still significantly longer than existing LIC methods.
>
> Our aim is to investigate the potential of diffusion models for compression purposes, for now focusing on fundamental rate-distortion and rate-perception tradeoffs. This is in line with the general trend in the neural compression community, where all existing approaches are currently 1-2 orders of magnitude slower than conventional image codecs (a recent exception is [Yang et al. 2023](arxiv.org/pdf/2304.06244.pdf)). We thus stress that our achieved decoding time performance is not completely unreasonable and in line with existing work. We remain optimistic that, through the development of enhanced architectures and continued research, runtime competitiveness can be substantially improved in the future. We believe that the current limitations in speed should not serve as a basis for dismissing early-stage research in neural compression.
>
> > I prefer reporting comparison results [...] on only 2-3 representative [...] metrics
>
> We appreciate the suggestion of reducing the evaluation metrics in the main paper, but since the most important perceptual metrics are still subject to an ongoing debate, we prefer to keep eight (out of 16) metrics in the main paper. All 16 metrics were reported in Appendix H. We also provide a brief discussion of the metrics we used in Appendix D.
>
> > I am confused about the theoretical analysis or rationale behind why using the X-parameterization..
>
> Please see our general response 3.
>
> > Code release
>
> Please see our general response 4.

---

> > ### Comment · Reviewer_kiiX · 2023-08-18
> >
> > Thanks for the responses from the authors. Concerning the paper itself and the comments by other reviewers, I will stick to my original rating.

---

### Author Rebuttal · Authors · 2023-08-09

# General Response:

We thank all reviewers for their valuable feedback. Before we address them individually, we would like to address common questions in their reviews.

> 1. Motivation, novelty, and advantages of our work

Our diffusion model provides a new approach towards “perceptual compression”, a field of growing importance (See Yang et al. 2022 https://arxiv.org/pdf/2202.06533.pdf). All current SOTA VAE-based neural image compression methods involve a probabilistic decoder p(x|z) that models the distribution over possible image reconstructions x given a global latent image representation z. While VAE-based decoders typically reconstruct the *mean* of p(x|z), this can lead to blurry artifacts since the true p(x|z) is multi-modal and hence the mean can be far away from the data manifold. In contrast, our approach *samples* from p(x|z) using a conditional diffusion model. This avoids mode averaging, leading to better perceptual qualities. Thus, our approach is a first alternative to an important class of GAN-based approaches (such as HiFiC) that bias the data reconstruction deterministically towards the mode of p(x|z). Such models of perceptual compression are currently at the forefront of learned image compression due to their potential for extreme compression gains.

We show empirically that our approach outperforms GAN-based approaches on various perceptual metrics as well as traditional distortion metrics. We also show that diffusion models allow various decoding strategies with one trained model, which provide more decoding flexibility than GAN-based models.

Our work exhibits novelty in several aspects. It is the first successful application of transform coding with diffusion models. Other diffusion approaches either focus on lossless compression (which is fundamentally different) or rely on more computationally expensive relative entropy coding schemes, as we have already discussed in our paper.

Additionally, our approach involves several crucial design decisions that contribute to its strong performance:
X-prediction: This technique enhances the reconstruction quality in our model, enabling stochastic image reconstructions in only a handful (as opposed to hundreds) of iterations.
Conditioning mechanism: Our unique method for incorporating a global image representation ensures competitive diffusion model performance.
An additional optional LPIPS loss can boost the perceptual quality of the diffusion model.

> 2. Why our diffusion-based approach uses an LPIPS loss

We stress that the LPIPS term is an optional addition in both HiFiC (an important perceptual compression baseline) and in our approach. HiFiC is essentially a VAE-GAN hybrid model, where a GAN discriminator biases the VAE reconstructions towards a solution closer to the data manifold. In contrast, our diffusion model can be seen as a VAE-Diffusion hybrid model with a stochastic, diffusion-based decoder. From this perspective, it is not surprising that addition an additional loss such as LPIPS can improve the performance in *both* approaches, especially when the focus is on perceptual metrics. A model with three losses is just more expressive than a model with two. We also note that the LPIPS term is one way of modeling an additional data likelihood $p(x_0|x_n)$ in diffusion models. This likelihood term was previously shown to improve sample quality in other tasks with diffusion models, e.g., [Austin et al. 2021](https://arxiv.org/pdf/2107.03006.pdf) equation 5.

> 3. The rationale behind x-prediction

X-prediction was initially introduced in the diffusion model by [Sohl-Dickstein et al. 2015](http://proceedings.mlr.press/v37/sohl-dickstein15.pdf), who directly employed the variational lower bound (with x-prediction) as their training objective. Subsequently, the DDPM paper by [Ho et al. 2020](https://arxiv.org/pdf/2006.11239.pdf) further established the connection between the variational lower bound and the $\epsilon$-prediction based score-matching loss. As demonstrated in the paper [Saliman et al. 2022](https://arxiv.org/pdf/2202.00512.pdf), empirical evidence has shown that x-parameterization can outperform epsilon-prediction, particularly when the number of sampling steps is limited (with progressive distillation) in unconditional generation.

Inspired by the above works, while we originally experimented with epsilon-parameterization, we found the surprising result that x-parameterization speeds up sampling by more than an order of magnitude. A possible explanation is that we utilize an encoder to generate a content latent variable, which is adept at providing a robust signal for input reconstruction even within a single denoising step (as illustrated in Fig 4), and that our loss resembles a VAE in this scenario. We note that our Eq.6 also described the mathematical equivalence between epsilon and x-prediction.

> 4. Appendix and Code availability

We note that we provided an appendix in the supplemental material. We apologize that we didn’t attach the necessary links or zip files regarding the code in our paper; this was originally intended by us but we lost track of it. According to the NeurIPS guidelines, we can not share a link to a repository at this stage, but we send the code to the Area Chair who can be contacted by you to obtain the code.

---

### Decision · Program_Chairs · 2023-09-21

**Decision:**

Accept (poster)

**Comment:**

Author rebuttal addressed most of the reviewer concerns and all reviewers are leaning accept after final discussion.